# Steering Generative Models with Experimental Data for Protein Fitness Optimization

**Jason Yang**[†]
Chemistry & Chemical Engineering
California Institute of Technology

**Wenda Chu**[†]
Computing & Mathematical Sciences
California Institute of Technology

**Daniel Khalil**
Computing & Mathematical Sciences
California Institute of Technology

**Raul Astudillo**
Computing & Mathematical Sciences
California Institute of Technology

**Bruce J. Wittmann**
Office of the Chief Scientific Officer
Microsoft Corporation

**Frances H. Arnold**
Chemistry & Chemical Engineering
Biology & Biological Engineering
California Institute of Technology

**Yisong Yue**[*]
Computing & Mathematical Sciences
California Institute of Technology

## Abstract

Protein fitness optimization involves finding a protein sequence that maximizes desired quantitative properties in a combinatorially large design space of possible sequences. Recent advances in steering protein generative models (e.g., diffusion models and language models) with labeled data offer a promising approach. However, most previous studies have optimized surrogate rewards and/or utilized large amounts of labeled data for steering, making it unclear how well existing methods perform and compare to each other in real-world optimization campaigns where fitness is measured through low-throughput wet-lab assays. In this study, we explore fitness optimization using small amounts (hundreds) of labeled sequence-fitness pairs and comprehensively evaluate strategies such as classifier guidance and posterior sampling for guiding generation from different discrete diffusion models of protein sequences. We also demonstrate how guidance can be integrated into adaptive sequence selection akin to Thompson sampling in Bayesian optimization, showing that plug-and-play guidance strategies offer advantages over alternatives such as reinforcement learning with protein language models. Overall, we provide practical insights into how to effectively steer modern generative models for next-generation protein fitness optimization.

## 1 Introduction

Proteins, sequences of amino acids, can be optimized for useful properties such as binding affinity, catalytic activity, or stability, numerically quantified as "fitness." However, protein optimization is

---

[*]yyue@caltech.edu
[†]These authors contributed equally to this work.

39th Conference on Neural Information Processing Systems (NeurIPS 2025).

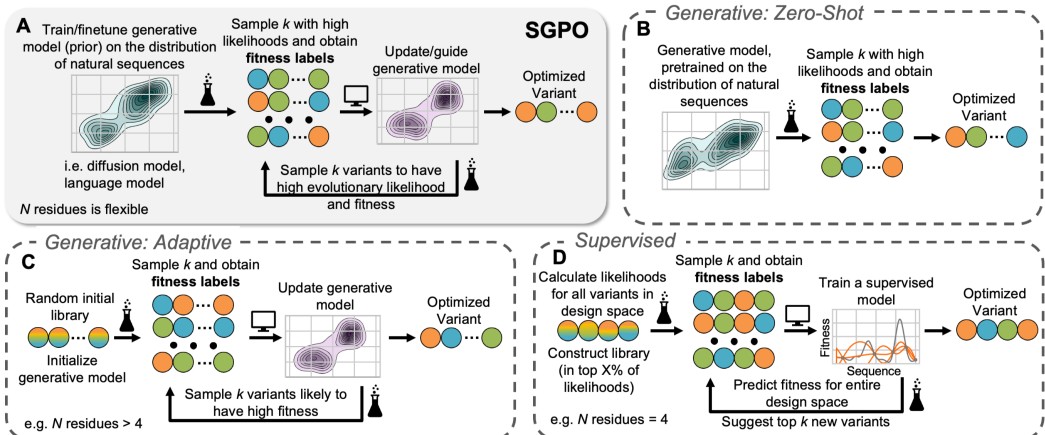

Figure 1: **Comparison of steered generation for protein optimization (SGPO) to other ML-assisted workflows for protein engineering. (A)** SGPO involves initializing a generative prior model to sample sequences with high natural likelihoods and steering that model with assay-labeled fitness data. Optimization is difficult because the design space is vast, and the throughput of wet-lab fitness assays (Erlenmeyer flask icon) is low, so adaptive learning across multiple iterations is beneficial. Previous methods have utilized generative models such as **(B)** fully zero-shot methods that sample highly natural sequences but do not utilize labeled fitness data or **(C)** those that only utilize labeled fitness. **(D)** Alternatively, supervised approaches involve enumerating to calculate fitness predictions for all variants in a design space, limiting them to optimizing few residues (i.e., $N < 9$).

challenging: the design space of proteins is enormous, as a protein of length $M$ can be constructed in $20^M$ different ways, of which only a negligible fraction are functional (Romero & Arnold, 2009). Moreover, most wet-lab assays only provide $10^2 - 10^3$ fitness labels. Consequently, researchers often rely on directed evolution, an iterative process aiming to incrementally improve protein fitness (Packer & Liu, 2015) through multiple rounds of mutation and experimental screening. In each round, a protein is mutated, the variants' fitnesses are measured, and the most beneficial variant is selected for the next iteration. However, this approach can be slow, often accumulating only one mutation per round, and inefficient, as it performs a local search limited to closely related protein sequences.

In recent years, there has been a growing interest in developing machine learning (ML)-assisted methods to optimize protein fitness more efficiently (Yang et al., 2019; Wittmann et al., 2021a; Hie & Yang, 2022; Yang et al., 2024, 2025c). Many recent studies have focused on generative approaches combining unlabeled and labeled data for protein design. Broadly, these methods achieve *conditional generation* by steering generative priors of natural protein sequences (Freschlin et al., 2022) using fitness data, thereby enabling incorporation of the steered models into adaptive optimization cycles (Hie & Yang, 2022). We refer to this class of methods as **S**teered **G**eneration for **P**rotein **O**ptimization (SGPO). These methods address the individual limitations of previous approaches (Fig. 1A, Table 1). First, SGPO leverages labeled data, which is essential for fitness goals that deviate from natural functions (e.g., engineering enzymes for non-native activities (Arnold, 2018; Yang et al., 2025b)), unlike zero-shot methods relying solely on generative priors of natural sequences (*Generative: Zero-Shot*, Fig. 1B, Hie et al. 2023; Sumida et al. 2024; Fei et al. 2025; Seki et al. 2025; Lambert et al. 2025). Second, generative priors (Wu et al., 2021; Hsu et al., 2024) sample sequences with high evolutionary likelihoods and potentially higher fitness, giving these methods a significant advantage over approaches relying exclusively on labeled data (Song & Li, 2023; Stanton et al., 2022; Gupta & Zou, 2019; Brookes & Listgarten, 2020; Jain et al., 2022; Kim et al., 2025; Angermueller et al., 2020; Hie & Yang, 2022) (*Generative: Adaptive*, Fig. 1C). Finally, SGPO scales to larger design spaces, unlike most supervised ML-assisted directed evolution (MLDE) approaches, which require enumerating and scoring all variants in the design space (*Supervised*, Fig. 1D) (Wu et al., 2019; Wittmann et al., 2021b; Yang et al., 2025b; Li et al., 2025a; Vornholt et al., 2024; Jiang et al., 2024; Hsu et al., 2022; Ding et al., 2024; Hawkins-Hooker et al., 2024; Zhao et al., 2024a; Thomas et al., 2025; Sun et al., 2025).

Despite these advantages, SGPO methods still face practical limitations in real-world fitness optimization, particularly across two major classes of approaches: guiding discrete diffusion models (Nisonoff et al., 2025; Stark et al., 2024; Klarner et al., 2024; Gruver et al., 2023; Lisanza et al.,

Table 1: **SGPO is a general approach for protein fitness optimization that does not face the individual limitations of other strategies.** Namely, SGPO utilizes zero-shot knowledge from the natural distribution of proteins, can be guided by assay-labeled fitness data, and can optimize many residues ($N$) simultaneously. Beyond those listed here, there are many other studies that combine different elements of these approaches.

| Approach | Prior Information Used? | Assay Fitness Used? | Scales to large $N$? | Protein Examples (non-exhaustive) |
|---|---|---|---|---|
| **SGPO** | ✓ | ✓ | ✓ | Lisanza et al. (2025); Widatalla et al. (2024); Stocco et al. (2024); Nisonoff et al. (2025); Brookes et al. (2019); Blalock et al. (2025); Goel et al. (2025); Huang et al. (2025) |
| Generative: Zero-Shot | ✓ | × | ✓ | Hie et al. (2023); Sumida et al. (2024); Fei et al. (2025); Seki et al. (2025); Lambert et al. (2025) |
| Generative: Adaptive | × | ✓ | ✓ | Song & Li (2023); Jain et al. (2022); Angermueller et al. (2020); Stanton et al. (2022); Brookes & Listgarten (2020) |
| Supervised | ✓ | ✓ | × | Wittmann et al. (2021b); Ding et al. (2024); Hawkins-Hooker et al. (2024); Zhao et al. (2024a); Sun et al. (2025) |

2025; Goel et al., 2025) and finetuning models such as protein language models (PLMs) through reinforcement learning (RL) (Ruffolo & Madani, 2024; Widatalla et al., 2024; Stocco et al., 2024; Blalock et al., 2025; Wang et al., 2025c). The limitations of prior work are summarized as follows: (1) Few previous studies have explored steering with few ($10^2 - 10^3$) labeled sequences (Lisanza et al., 2025; Stocco et al., 2024) for protein optimization based on real fitness data, e.g. activity or fluorescence, rather than computational surrogates (Lisanza et al., 2025; Blalock et al., 2025). (2) Most studies only evaluate one type of generative prior and steering strategy, so it is unclear how different combinations perform in practice. (3) There is room to incorporate principles from adaptive optimization, such as uncertainty-aware exploration (e.g., Bayesian optimization), which have shown clear benefits in protein engineering (Vornholt et al., 2024; Yang et al., 2025b).

In this study, we aim to understand the best practices for integrating SGPO into real-world engineering workflows. We focus here on modern generative models (i.e., discrete diffusion, language models) but acknowledge that other related methods are relevant, such as those based on variational autoencoders (Brookes et al., 2019; Torres et al., 2024) and other adaptive search strategies (Kirjner et al., 2024; Sinai et al., 2020; Ren et al., 2022). We explore the following questions: Which steering strategies perform best, and with which types of models? How can we utilize uncertainty to better explore the design space when performing guidance? **Overall, we make the following key contributions:**

1. We motivate SGPO as a useful, general framework and contextualize existing methods for protein optimization under this umbrella.

2. We comprehensively evaluate design decisions for SGPO, including different generative models for sequences and steering strategies (Fig. 2 & 3, Section 2), offering best practices for protein optimization with few fitness labels.

3. We introduce ideas from adaptive optimization into SGPO by proposing a method that ensembles multiple plug-and-play fitness predictors and leverages their predictive uncertainty to enable more efficient exploration.

4. We are the first to adapt *decoupled annealing posterior sampling* (Zhang et al., 2025) for SGPO, and this type of plug-and-play guidance has the strongest performance overall.

On the TrpB, CreiLOV, and GB1 protein fitness datasets, we find that SGPO methods can consistently identify high-fitness protein variants. In particular, our results highlight the advantages of plug-and-play guidance with diffusion models over finetuned language models—offering greater steerability and lower computational cost. To support future research and real-world adoption, our extensive, user-friendly code is available at `https://github.com/jsunn-y/SGPO`.

## 2 Related work

**Generative models for discrete sequences.** The most widely adopted generative models for natural protein sequences are PLMs, such as autoregressive transformers (Nijkamp et al., 2023) and masked language models (Rives et al., 2021). Increasingly, various diffusion model (Ho et al., 2020) architectures have shown efficacy for modeling discrete data ($\mathbf{x}$) (Li et al., 2025b), such as protein sequences (Alamdari et al., 2024; Wang et al., 2024), leveraging many similar learning techniques such as masking or autoregressive decoding (Sahoo et al., 2024; Lou et al., 2024; Nie et al., 2025; Shi et al., 2024) (Fig. 2). These generative prior models $p(\mathbf{x})$ can be categorized broadly into two types: those that perform diffusion in a continuous latent space (Li et al., 2022; Chen et al., 2023b; Dieleman et al., 2022) and those that diffuse directly over discrete space (Fig. 2). In the protein domain, it has also been shown that latent diffusion over embeddings from PLMs can be more effective (Meshchaninov et al., 2025; Chen et al., 2023a; Torres et al., 2025). Alternatively, models performing diffusion in discrete space use a transition matrix to update all discrete states in each timestep (D3PM) (Austin et al., 2021), which has later been formulated as continuous-time Markov chains (Lou et al., 2024; Campbell et al., 2022, 2024; Schiff et al., 2024). Two common ways to add noise to discrete sequences are to use uniform noise matrices or absorbing state (masking) matrices (Fig. 2). These have been followed by simplified frameworks showing some of the highest performance for modeling natural language, such as masked diffusion language models (MDLMs) (Sahoo et al., 2024; Hoogeboom et al., 2022; Shi et al., 2024) and a variation that uses uniform noise called uniform diffusion language models (UDLMs) (Schiff et al., 2024). We elaborate more on these methods in Section A.3.

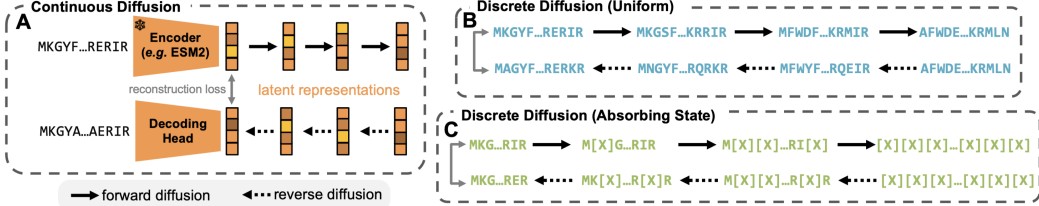

Figure 2: **Overview of different approaches to train diffusion models over discrete state spaces.** During inference, a noised latent representation or sequence is decoded into a reasonable sequence (bottom track for each method). [X] refers to a masked token.

**Plug-and-play guidance strategies.** An advantage of diffusion models is the ability to perform plug-and-play guidance based on fitness labels ($\mathbf{y}$) without finetuning the generative prior model weights, resulting in reduced training costs and potentially strong signal despite having few ($\sim 10^2$) labels. Guiding a continuous diffusion model often involves skewing the learned score function using gradients from a supervised value function that can predict labels $\mathbf{y}$ from data $\mathbf{x}$ (Chung et al., 2023; Zheng et al., 2025; Soares et al., 2025). These methods are often referred to as posterior sampling, as they aim to sample from the posterior distribution, $p(\mathbf{x}|\mathbf{y})$. Recent works extend this idea to guiding discrete diffusion models. Classifier guidance (CG) (Nisonoff et al., 2025) skews the rate matrix of the reverse time Markov chain of discrete diffusion models using a time-dependent value function, $p(\mathbf{y}|\mathbf{x}_t, t)$; variable splitting methods (DAPS) (Zhang et al., 2025; Chu et al., 2025) use discrete diffusion models as denoisers and only require a value function of clean data, $p(\mathbf{y}|\mathbf{x}_0)$; diffusion optimized sampling (NOS) (Gruver et al., 2023) trains a value function on continuous embeddings of discrete tokens and optimizes the embedding for higher fitness; sequential Monte Carlo methods (SMC) (Li et al., 2024a; Uehara et al., 2025; Wu et al., 2024; Lee et al., 2025a; Singhal et al., 2025) evolve multiple particles from a series of distributions to approximate the posterior distribution in limit. We explain these methods in more detail in Section A.4, along with other variations on the guidance process. In this study, we focus on CG, DAPS, and NOS as guidance techniques (Fig. 3). Future work could also consider guidance techniques for autoregressive language models, such as future discriminators for generation (FUDGE) (Yang & Klein, 2021), plug and play language models (PPLM) (Dathathri et al., 2020), and twisted SMC (Zhao et al., 2024b; Amin et al., 2025b). Additionally, Xiong et al. (2025) demonstrate how guidance generalizes to masked language models and order-agnostic autoregressive models.

**Reinforcement learning via model finetuning.** We consider RL broadly here as techniques that achieve conditional generation by finetuning generative models with labeled data, thus pushing

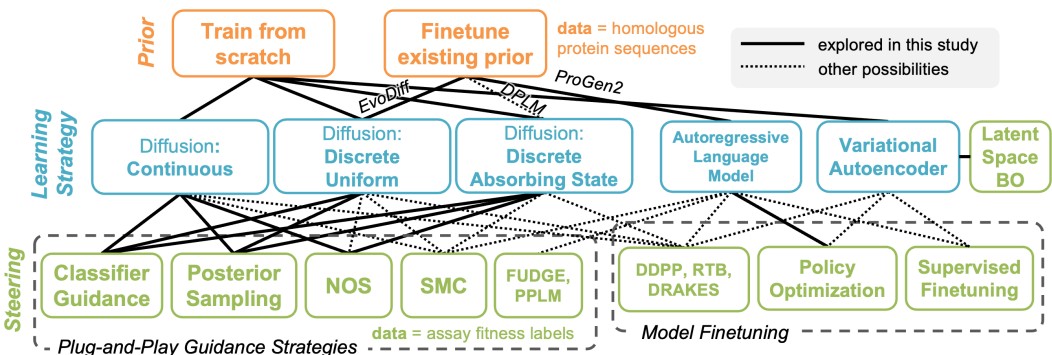

Figure 3: **Methods design space for SGPO: a non-exhaustive landscape of generative models for protein *sequences* and methods to steer them with labeled data.** Three major types of diffusion models for sequences include those that perform diffusion over continuous space and those that perform diffusion over discrete space with a uniform or absorbing state (masking) noising process. Various types of guidance strategies are compatible with certain models, in green (NOS: diffusion optimization sampling, SMC: sequential monte carlo, FUDGE: future discriminators for generation, PPLM: plug and play language models, DDPP: discrete denoising posterior prediction, RTB: relative trajectory balance, DPLM: diffusion protein language model, BO: Bayesian optimization). Differently, language models and variational autoencoders can be aligned with labeled data via reinforcement learning such as policy optimization or supervised finetuning.

those models to produce more favorable generations. There are emerging RL techniques applied to discrete diffusion models, including discrete denoising posterior prediction (DDPP) (Rector-Brooks et al., 2025), relative trajectory balance (RTB) (Venkatraman et al., 2024; Bartoldson et al., 2025; Venkatraman et al., 2025), and direct reward backpropagation with gumbel softmax trick (DRAKES) (Wang et al., 2025a). While the above strategies are specific to discrete diffusion models, supervised fine-tuning (SFT) and policy optimization are two important techniques used in RL that can be broadly applied to generative models such as language models (Fig. 3). Policy optimization has generally shown better performance than SFT (Stocco et al., 2024; Blalock et al., 2025); in particular, direct preference optimization (DPO) is often used for its algorithmic simplicity and ease of training (Rafailov et al., 2023) (details in Section A.4). RL has demonstrated utility for aligning generative models of proteins (language models, inverse folding models, variational autoencoders) with properties like stability (Widatalla et al., 2024; Blalock et al., 2025; Stocco et al., 2024; Lim et al., 2025), but these methods can have high computational costs of finetuning and may require large amounts of labels ($> 10^3$) to effectively steer generations. We include DPO with an autoregressive PLM (finetuned ProGen2 (Nijkamp et al., 2023)) as a baseline.

**Adaptive optimization.** Protein engineering is commonly conducted through adaptive workflows such as directed evolution Packer & Liu (2015) or ML-based approaches such as Bayesian optimization (Frazier, 2018; Stanton et al., 2022). These methods follow an iterative loop: labeled data is collected via expensive wet-lab assays, a surrogate model $p(\mathbf{y}|\mathbf{x})$ is trained or updated, an acquisition function implied by the surrogate is used to propose new sequences to evaluate, and the cycle repeats (Hie & Yang, 2022; Vornholt et al., 2024; Yang et al., 2025b). The surrogate model, often a Gaussian process or a deep ensemble, provides uncertainty estimates, which are used by an acquisition function (e.g., expected improvement, Thompson sampling) to balance exploration and exploitation of the design space. In this study, we adapt these ideas to guide diffusion models for protein sequence generation, as described in Section 4.3. A closely related line of work is latent space Bayesian optimization (Maus et al., 2022; Stanton et al., 2022; Gómez-Bombarelli et al., 2018; Castro et al., 2022; Torres et al., 2024; Lee et al., 2025b), which searches for optimal sequences within a latent space—typically learned by an autoencoder, which can implicitly capture a prior on natural protein sequences. In this work, we compare against APEXGo (Torres et al., 2024), a method that performs trust-region Bayesian optimization in the latent space of a variational autoencoder trained over protein sequences. There are also related methods that involve conditional sampling from a prior (Brookes et al., 2019). However, we note that SGPO offers greater flexibility by avoiding reliance on an explicit latent space, which enables the use of modern, more powerful generative models such as diffusion models and protein language models that are not easily accommodated by traditional latent Bayesian optimization pipelines.

# 3 Problem setup

We focus on evaluating methods that fall under SGPO, where the primary downstream task entails starting from a known sequence with some level of fitness for a target objective (i.e. activity, stability, fluorescence, binding, etc.) and identifying a modified sequence with maximized fitness, where real-world fitness can only be measured for $10^2$ to $10^3$ sequences. Our goal is to sample sequences with maximum fitness $\mathbf{y}$ from the *generative prior* $p(\mathbf{x})$, which is trained on the multiple sequence alignment (MSA) of homologous protein sequences that are evolutionarily related to a known protein with some level of desired fitness (details in Section A.3). This model can be thought of as capturing the distribution of sequences with high likelihood from a given protein family.

During inference, sequences can be sampled *unconditionally* from $p(\mathbf{x})$, or sampling can be *guided* using a supervised model of the form $p(\mathbf{y}|\mathbf{x}) \propto \exp(f(\mathbf{x})/\beta)$, where $f(\cdot)$ is a learned fitness predictor—also referred to as the *classifier* or *value function*. This predictor is trained on a small number of labeled sequence-fitness pairs (typically in the hundreds) to reflect practical data limitations. The goal of guided sampling is to generate protein sequences from the posterior distribution, $p(\mathbf{x}|\mathbf{y}) \propto p(\mathbf{x}) \exp(f(\mathbf{x})/\beta)$. We use a computational *oracle* to acquire and evaluate fitness labels $\mathbf{y}$, to simulate how fitness would be measured in a real-world campaign. Details on training and guidance with the value function are provided in Section A.4 and Table A2. As an alternative steering method to guidance, we finetune the generative prior with labeled data using an autoregressive language model (ARLM) and DPO, which serves as a baseline. We further compare to a baseline of latent space Bayesian optimization. The strength of steering is tuned by method-specific hyperparameters.

# 4 Results

Table 2: **Summary of datasets used in this work.** Train and test fitness refer to the number of fitness labels used for training and testing the oracle. We focus on TrpB and CreiLOV, with some of the GB1 results moved to the Appendix. While the TrpB dataset has a lot more training labels, it may be more difficult to learn due to relatively high amounts of epistatic effects between residues (non-additivity of mutation effects).

| Dataset | Length | Targeted Residues | Design Space | MSA Size | Train Fitness | Test Fitness | Reference |
|---|---|---|---|---|---|---|---|
| **TrpB** Enzyme Activity | 389 | 117, 118, 119, 162, 166, 182, 183, 184, 185, 186, 227, 228, 230, 231, 301 | $N$=15 | 5.7$e$4 | 75,618 | 23,313 | Johnston et al. (2024) |
| **CreiLOV** Fluorescence | 119 | All | $N$=119 | 3.7$e$5 | 6,842 | 2,401 | Chen et al. (2023c) |
| **GB1** Binding | 56 | All | $N$=56 | 126 | 3.9e6 | 9.6e4 | Olson et al. (2014) |

We study three proteins, the TrpB enzyme (Johnston et al., 2024), the CreiLOV fluorescent protein (Chen et al., 2023c), and the GB1 binding protein (Olson et al., 2014) due to the availability of fitness data across many residues (Table 2). We focus protein fitness optimization to a design space of 15 residues in TrpB (only these positions are allowed to vary) and all 119 and 56 residues in CreiLOV and GB1, respectively. For each protein's variants, we evaluate fitness by approximating it via a supervised oracle trained on a large amount of real data (Section A.2).

## 4.1 Model pretraining captures the distribution of evolutionarily related protein sequences and enables sampling sequences with high fitness

Based on the methods explained in Section A.1 and A.3, we trained generative priors on natural sequences from the MSA, focusing on continuous diffusion models (*Continuous*), discrete diffusion models with uniform (*D3PM, UDLM*) and absorbing state noising processes (*MDLM*), and autoregressive language models (*ARLM*) (Table 3). Overall, the trained models capture the natural distribution of protein sequences, with the D3PM models seeming to match the distribution the most

Table 3: **Summary of generative priors evaluated in this work.** Each generative prior was trained on an MSA of homologous natural sequences. All denoising processes were modeled using a transformer architecture (Section A.3). *Italicized* models were further explored in downstream guidance experiments.

| Model | Type | Noise | # Params | Notes |
|---|---|---|---|---|
| ***Continuous*** Continuous-ESM | Continuous Diffusion | Gaussian | 27.9 M 25.5 M | diffusion over ESM embeddings |
| D3PM-Baseline ***D3PM*** UDLM | Discrete Diffusion | Uniform | 37.9 M 37.9 M 28.6 M | finetuned from EvoDiff 38M-Uniform (Alamdari et al., 2024) uniform diffusion langauge model |
| ***MDLM*** | Discrete Diffusion | Absorbing | 28.6 M | masked diffusion language model |
| ***ARLM*** | Language Model | n/a | 151 M | autoregressive language model finetuned from ProGen2-small (Nijkamp et al., 2023) |

closely while also generating sequences with high diversity (Fig. 4, Fig. A3). The two different diffusion models over continuous space show comparatively lower performance, and diffusing over the latent space of ESM embeddings does not boost performance on this task. The UDLM model has low performance due to mode collapse (Fig. A3, Fig. A5). Future work could finetune the pretrained diffusion protein language model (DPLM) as an MDLM (Wang et al., 2024).

Overall, we found that pretrained priors sample protein variants that have higher mean fitness, which corroborates previous studies finding that sequences with higher evolutionary likelihood are also likely to have higher fitness (Li et al., 2025a; Hie et al., 2023). Based on these results, we proceeded to perform remaining experiments with one model from each category of model type, namely the *Continuous*, *D3PM*, *MDLM*, and *ARLM* models.

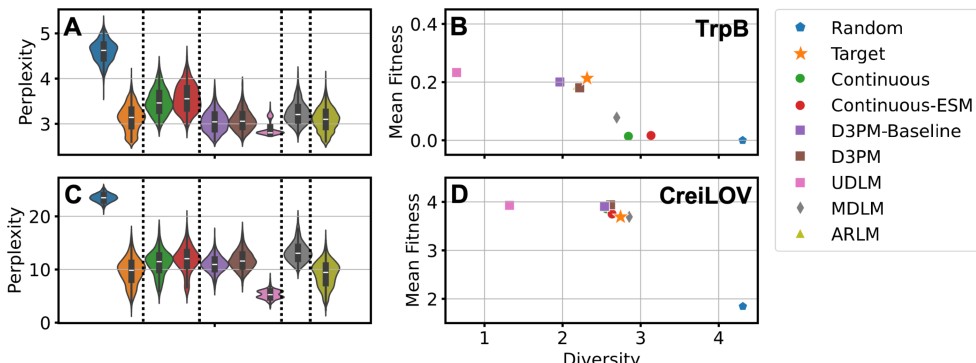

Figure 4: **Pretrained generative priors capture the target distribution of naturally occurring sequences** that are homologous to TrpB (**A-B**) and CreiLOV (**C-D**), respectively. Lower perplexity corresponds to higher likelihood in the model. The diversity of sequences was computed as the average Shannon entropy of mutated positions with mean fitness corresponding to the oracle predictions. While the various models largely achieve comparable performance, the D3PM models capture the target distribution with the highest fidelity, whereas the UDLM model is prone to mode collapse. For each model, 1000 sequences were sampled and repeats were allowed to approximate the distribution. To approximate the target distribution, 1000 sequences were sampled from the MSA used for pretraining. Perplexity was calculated by passing generated sequences through the 764 M parameter ProGen2-base model. More details on model training can be found in Table 3 and Section A.3, and GB1 results are provided in Fig. A4.

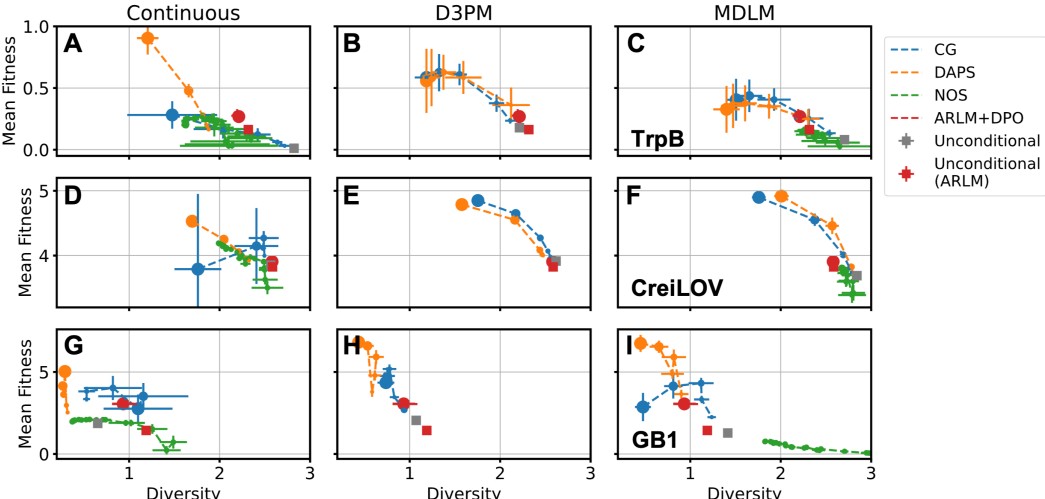

Figure 5: **Pareto boundaries demonstrate the trade-off between generating sequences with high fitness and high diversity** for TrpB (**A-C**), CreiLOV (**D-F**), and GB1 (**G-I**). Sequences sampled from the generative models (Continuous, D3PM, and MDLM), after guidance with labeled fitness data, are enriched in high-fitness protein variants, and most methods show higher performance than the ARLM+DPO baseline. Larger circle indicates a stronger guidance strength hyperparameter (excluding NOS), specified in Table A3. Each experiment was repeated using 10 different standardized sets of 200 unique sequences used for steering, each drawn from the D3PM prior, and error bars show standard deviation. Mean fitness and diversity were calculated based on 200 generated samples, with diversity calculated as the average Shannon entropy of amino acids at mutated positions. Unconditional refers to sequences sampled from the prior with no guidance.

## 4.2 Evaluating SGPO design choices

Impressively, steering with modest amounts of labeled data (200 sequence-fitness pairs) enables most models and methods to generate sequences with even higher fitness, while sacrificing some generation diversity (Fig. 5). In this low data regime, guidance with diffusion models outperforms DPO with language models; the latter does not enable as much steerability. CG and DAPS enable the strongest steerability overall, but DAPS outperforms CG for the continuous models (Fig. 5A, D). In general, guidance seems to work similarly for uniform diffusion (D3PM) and to absorbing state diffusion (MDLM). Overall, the continuous diffusion models do not perform as well as other models, as the prior does not capture the distribution of natural sequences with high fitness as well (Fig. 5A). NOS does not seem to allow for as much steerability, despite an extensive hyperparameter scan (Table A3). Finally, we conducted a closer analysis of the number of unique sequences generated by the steered models and confirmed that most models produce entirely novel sequences, suggesting that they are not over-steering (Fig. A6).

## 4.3 "Thompson sampling" using an ensemble of classifiers is effective for adaptive optimization

Next, we performed adaptive optimization experiments, which mimic real-world protein engineering scenarios and follow a setup similar to batch Bayesian optimization: in each round, a batch of sequences is sampled, evaluated for fitness, and used to retrain a supervised value function that guides sampling from the pretrained prior. We focused on the MDLM models with the CG and DAPS guidance strategies, as these combinations achieved the best performance in our earlier set of experiments (Fig. 5). Based on findings from these previous experiments, we selected the ideal guidance strength hyperparameter to balance fitness and diversity—ensuring high predicted fitness without significantly compromising sequence diversity (Table A3). For both guidance strategies, we employed an algorithm akin to Thompson sampling (Kandasamy et al., 2018; Russo et al., 2018), drawing a different value function from a frequentist ensemble of neural network regressors to guide the generation of each new sample (Yang et al., 2025b). Pseudocode for our adaptive optimization algorithm is provided in Section A.5.

Plug-and-play guidance strategies outperform baselines such as DPO with an ARLM, sampling just from the unconditional generative prior, and latent space Bayesian optimization with APEXGo (Fig. 6): Sampled sequences achieve higher values of mean and maximum fitness. Furthermore, campaigns using an ensemble of value functions and "Thompson sampling" achieve higher maximum fitness than those using only a single value function for guidance (Table A4), which may be because these models enable more exploration of sequence space (Fig. A7). However, it is difficult to ascertain wither CG or DAPS works better as a guidance strategy, as the performance is highly dependent on the guidance strength hyperparameter, and the optimal hyperparameter will not typically be known in a real-world campaign. Because the oracle may not capture the true nature of the protein fitness landscape, we also suggest making relative comparisons here rather than absolute comparisons between model performance.

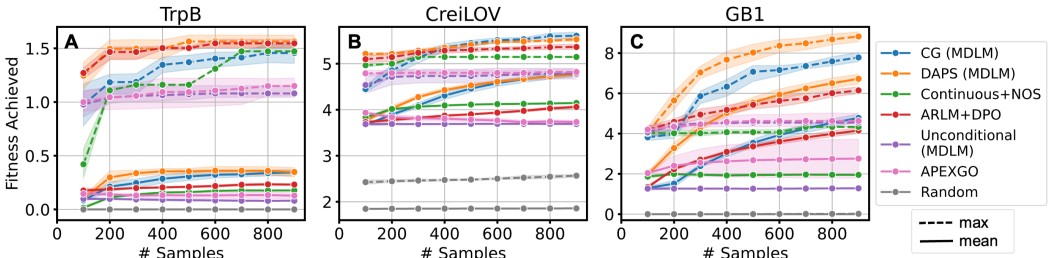

Figure 6: **Maximum/mean fitness achieved improves over multiple iterations of steering in an adaptive setting similar to batch Bayesian optimization** for TrpB (**A**), CreiLOV (**B**), and GB1 (**C**). 100 sequences were sampled in each round. Within each round, an ensemble of 10 value functions (classifiers) was trained on fitness data from all previously queried samples, and each new sample was generated by the MDLM model guided with a value function sampled from the ensemble (akin to Thompson sampling). Only unique, novel samples were acquired. Guidance strength parameter is provided in Table A3. Error bars show standard deviation between 5 different random initializations.

## 5 Discussion

In this work, we conduct a comprehensive study of SGPO methods and demonstrate that it is an effective approach for protein fitness optimization, by capturing the distribution of natural protein sequences with a generative prior and then steering the generations with labeled data. We find that DAPS with discrete diffusion models has the highest performance overall, and plug-and-play guidance-based strategies are generally more effective than finetuning language models; the latter can be difficult when only few fitness labels are available. SGPO approaches also outperform latent space Bayesian optimization (namely APEXGo), which we attribute to the difficulty in calibrating the trust region in very low-data regimes with limited rounds of optimization and the fact that latent space Bayesian optimization relies heavily on the structure of the latent space learned during generation model training, which can limit extrapolation to high-fitness but unnatural variants.

Using plug-and-play guidance approaches has other advantages. First, only one hyperparameter (guidance strength) needs to be tuned. In real-world engineering scenarios, even in the absence of ground truth fitness labels, one practical approach to selecting the guidance strength is to scan over values and choose the highest setting for which $n$ generated sequences remain unique and novel relative to previously measured sequences, where $n$ corresponds to the screening throughput available for the next round. By contrast, for DPO, various hyperparameters need to be tuned, and the training process has to be monitored closely. Even for NOS, different parameters such as the step size, the number of steps, and the stability coefficient must be tuned together. A further advantage of guidance is the low computational cost required, as the prior model weights are not updated during guidance. Pretraining/finetuning to obtain each initial prior was achieved on a single H100 GPU in less than one hour while each individual guidance experiment took minutes; pretraining language models took several hours on a single GPU.

Still, there are certain limitations of our work. We focused on proteins with fitness as mostly native function, but it would be interesting to test SGPO on other protein fitness optimization tasks where the pretrained prior may not provide as much utility. We also focused on protein optimization where

only $\approx 10^2$ fitness labels were available; different methods, such as RL, may perform better for applications where larger amounts of fitness data are available (Hie & Yang, 2022; Blalock et al., 2025). We focused on guidance strategies and did not test DPO or model finetuning-based methods with discrete diffusion models, but future work could adapt these methods for discrete diffusion (Borso et al., 2025). Furthermore, for TrpB and for language models, we manually mapped sequences back into the design space after generation (Section A.2), but explicitly building this into sampling techniques, such as inpainting in masked models (Blalock et al., 2025; Goel et al., 2025) may lead to improved performance. We did not consider insertions or deletions, but variable-length sequence generation could be considered in the future. Finally, we did not directly compare to existing approaches for protein engineering such as directed evolution for reasons explained in Section A.2.

There are several promising directions for future work to improve and extend SGPO methods. For instance, we experimented with guiding generation using value functions sampled from a Gaussian process posterior, enabling principled Thompson sampling from a fully Bayesian perspective. However, the Gaussian process struggled to model high-dimensional protein representations, leading to poor performance. This limitation could potentially be addressed with better kernel choices (Wilson et al., 2016; Michael et al., 2024; Yang et al., 2025b). Recent work has also begun to incorporate multi-objective optimization (Annadani et al., 2025; Tang et al., 2025a; Li et al., 2024b; Chen et al., 2025) and uncertainty quantification (Wu et al., 2025) when guiding diffusion models. Simultaneously, alternatives to acquisition-function-based approaches are being developed to enable Bayesian optimization in large design spaces where enumeration is infeasible (Bal et al., 2025). Other emerging approaches—closer in spirit to flow matching—are being proposed for discrete data and may offer new opportunities for exploration (Davis et al., 2024; Stark et al., 2024; Tang et al., 2025b). Finally, for masked diffusion models, strategies such as remasking or scheduling could be explored to improve inference, particularly to enhance model amenability to guidance (Wang et al., 2025b; Peng et al., 2025; Liu et al., 2025; Amin et al., 2025a). It will also be interesting to further explore guidance in other discrete domains such as natural language and small molecules (Schiff et al., 2024).

In summary, guiding generative models with labeled data offers a powerful, flexible, and principled framework for protein fitness optimization, as it effectively leverages both the evolutionary information encoded in natural protein sequences and task-specific fitness objectives. At the same time, we recognize the potential dual-use risks: such methods could, in principle, be misused to design harmful proteins, underscoring the importance of appropriate safeguards (Baker & Church, 2024; Wittmann et al., 2024). In short, our work has examined multiple effective SGPO strategies and offered insights on best-practices for real-world protein fitness optimization, laying the groundwork for further exploration and wet-lab validation.

**Acknowledgments**

This work was supported by a U.S. Army Research Office cooperative agreement (W911NF-19-2-0026 to F.H.A.) and an Amgen Chem-Bio Engineering award. J.Y. is also supported by the NSF Graduate Research Fellowship Program and the Google PhD Fellowship. We would like to thank Hunter Nisonoff, Jacob Gershon, Lucas Arnoldt, and Chenghao Liu for helpful discussions and Francesca-Zhoufan Li for help with the TrpB dataset. We would also like to thank Nate Gruver for help with the NOS implementation, Filippo Stocco for help with the DPO implementation, and Nathaniel Blalock for guidance on how to use the CreiLOV dataset.

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

# A  Appendix

## A.1  Data for pretraining generative priors

The first step in our pipeline involves learning a generative prior on naturally occurring protein sequences to capture the distribution of those with high evolutionary likelihood. This prior is unconditional in the sense that no labeled fitness data is used for training. However, because we are optimizing protein *variants* for a desired fitness, we pretrained our generative prior on sequences homologous to the parent protein to be optimized (known as a multiple sequence alignment or MSA): TrpB, CreiLOV, or GB1. Likelihoods from MSAs have been captured by statistical models and have been shown to offer good zero-shot approximations of fitness. In other words, they capture mutational substitutions that are more favorable, based on the precedent of natural evolution.

We focused on the TrpB (Johnston et al., 2024) and CreiLOV (Chen et al., 2023c) datasets due to the extensive number of sequences in their MSAs and compared to GB1 (Olson et al., 2014), which has comparatively fewer sequences. MSAs were obtained by running jackhmmer (Johnson et al., 2010) against Uniref90 for two iterations with the parent sequence of the fitness dataset as target. For the MSA, we only used sequences where the aligned portion was at least 75% the length of the parent sequence. We used the MSA that was aligned to the parent sequence, with gap tokens replaced by the corresponding amino acid found in the parent sequence, resulting in full, fixed-length pseudo-natural sequences. For GB1, we augmented the training set with synthetic data, namely all proteins with a single mutation to sequences in the MSA. For the language models on TrpB and CreiLOV, some sequences were randomly mutated by a single position near the beginning of the sequence, to prevent mode collapse during autoregressive generation.

We performed sequence clustering using mmseqs2 (Steinegger & Söding, 2017) at 80% identity and resampled the dataset by weighting each sample with $\frac{1}{1+\ln(n)}$ relative probability of being sampled, where $n$ is the size of the cluster associated with that sequence. Afterward, we removed 5% of the clusters and their associated sequences as a validation set.

## A.2  Protein fitness optimization task

**An oracle as a proxy for protein fitness.**    We studied fitness optimization across three different protein-fitness datasets, TrpB, CreiLOV, and GB1 (Table 2). TrpB is 389 residues in length, but based on available fitness data, we limited design to 15 residues: 117, 118, 119, 162, 166, 182, 183, 184, 185, 186, 227, 228, 230, 231, and 301. Namely, we combined the fitness data from 6 combinatorially complete 3-site libraries (D-I from Johnston et al. (2024)) and the 4-site library across residues 183, 184, 227, and 228. We normalized the parent fitness to 1 in each dataset and rounded all negative fitness values up to zero. The fitness here is the catalytic rate of a native reaction, the formation of tryptophan from indole and serine. To obtain a proxy fitness for all variants in the design space ($20^{15}$ possibilities) we trained an oracle inspired by the dataset splitting and model architecture used in Blalock et al. (2025). Namely, we used all of the single, double, and triple mutants in the library for training, with 10% and 20% of the quadruple mutants being used for validation and testing, respectively. Our model consists of an ensemble of 20 MLPs for TrpB, and each was trained on one-hot encodings of the designed residues for 1000 epochs.

Differently, the CreiLOV dataset (length $N = 119$) contains experimental fitnesses for all single mutations in the protein and certain higher order mutations at 15 selected positions with beneficial single mutations. Fitness here refers to associated fluorescence. To obtain a proxy fitness for all variants in the design space ($20^{119}$ possibilities), we trained an oracle similar to the procedure above, using similar splits to those in Blalock et al. (2025) and were able to reproduce their high performance on the test set. Before model training, we scaled the fitnesses of the single mutants to the fitnesses of multi-mutants by adding a normalization factor to all single mutants such that the parent sequence in both datasets had the same fitness. Our model consists of an ensemble of 10 MLPs for CreiLOV, and each was trained on onehot encodings of sequences for 1000 epochs.

For GB1, the experimental finesses for nearly all double mutations across the entire protein were available, where fitness refers to binding affinity of a domain of the G protein. To train the oracle, we held out 10% and 20% the sequences with two mutations as a validation and test set, respectively, with remaining sequences being used for training. Our model consists of an ensemble of 10 MLPs for GB1, and each was trained on one-hot encodings of the designed residues for 50 epochs.

Our oracles show high Pearson correlation on the train and test sets (Fig. A1). As the generalization ability of our oracle was only been tested on variants that are similar to the parent, we penalized the fitness of protein sequences by a factor of 0.99 for every mutation accumulated beyond a threshold of 60% sequence identity to the parent sequence. From here forth, we treated ground truth fitness as outputs from the oracle.

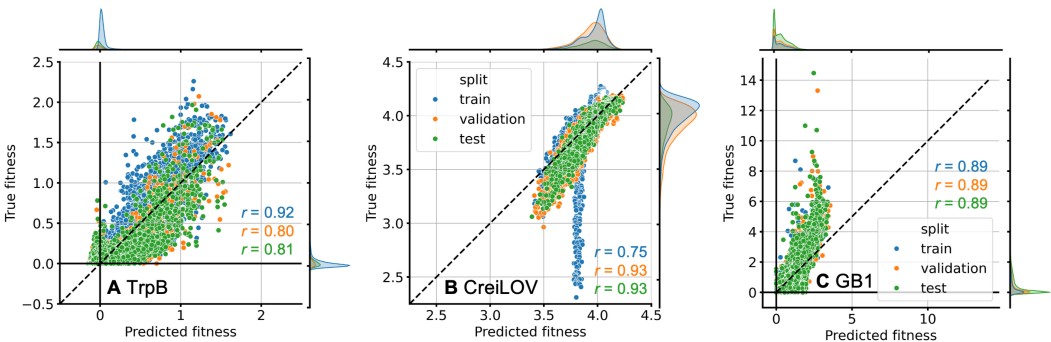

Figure A1: Oracles trained on available labeled fitness data for TrpB, CreiLOV, and GB1 extrapolate well to higher order combinations of mutations within the design space, as measured by Pearson correlation.

**Processing generated sequences.**    Our primary method for evaluation involved examining the distribution of sampled sequences and their corresponding fitness values, diversities, and novelties. The processing pipeline for generated sequences in shown in Fig. A2. In diffusion models, sequences were generated with fixed length equal to the parent length. For the language models, nearly all generated sequences had length equal to the parent sequence length. Still, sequences were aligned with the parent sequence using mafft (Katoh & Standley, 2013), and gaps were replaced with the corresponding amino acid in the parent sequence to generate complete pseudo-sequences of a fixed length. Special tokens, which occurred rarely in generation, were replaced by a random amino acid. For TrpB, residues outside of the design space of 15 residues were naively mapped to the original amino acid type in the parent sequence at the end of generation. We did not test inpainting, although this could be accomplished with masked (diffusion) language models.

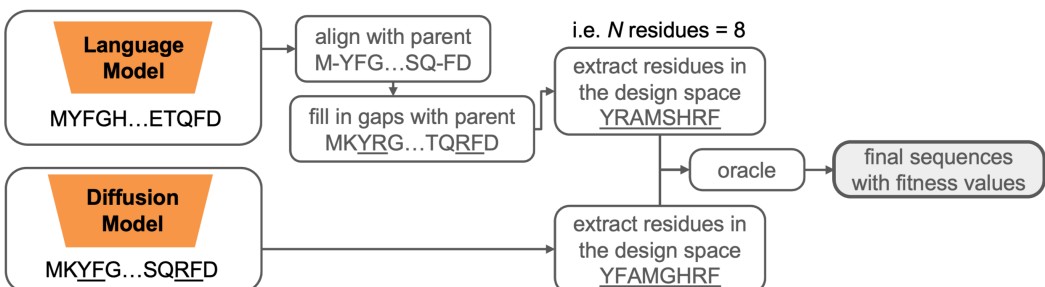

Figure A2: Example pipeline for generating protein sequences for evaluation, based on a hypothetical parent sequence: MKKFG...SQRFD (length=100), with 8 residues being optimized (3, 4, 26, 27, 28, 29, 98, 99), corresponding to a design space combo of KFDEACRF.

**Comparison to existing protein engineering methods.**    There are several reasons why we did not directly compare the performance of SGPO methods to existing methods used in protein engineering, such as directed evolution and MLDE. In the case of directed evolution (such as random mutagenesis): (1) It is not obvious which parent sequences to use as the starting points for directed evolution for a fair comparison. (2) It is unclear if the oracle captures the true nature of the protein fitness landscape or extrapolates well to sequences with many mutations relative to the original fitness dataset from which the oracle was trained. (3) Overall, our method enables the accumulation of many mutations in a single round of experimentation, whereas directed evolution is largely limited to one mutation at a time. For example, on the CreiLOV dataset, the generated sequences with the highest fitness had on average 66 mutations from the parent reference sequence from which the original dataset was

generated, which would not be achievable with directed evolution. We also did not directly compare our method to supervised approaches in smaller design spaces, such as 4-site combinatorial libraries (Yang et al., 2025b), as we focus here on design in larger design spaces, where existing methods are lacking. Overall, traversing large swaths of sequence space will be important for faster engineering and enabling improvements to fitness that would normally be slow with directed evolution.

## A.3 Generative models for sequences

Table A1: **Summary of training details for generative priors in this work.** Reference refers to the codebase that was modified for our implementation and where the model architecture was adapted from. For all models, we retained the model with the lowest validation loss. When using the ESM encoder, we used the 35M-parameter ESM2 model Lin et al. (2023).

| Model | Max Epochs | Learning Rate | Batch Size | Warmup Steps | Noise Schedule | Diffusion Timesteps | Model Architecture | Reference |
|---|---|---|---|---|---|---|---|---|
| **Continuous** | 5 | $1\times10^{-4}$ | 64 | 10 | cosine | 500 | BERT | Gruver et al. (2023) |
| Continuous-ESM | 25 | $1\times10^{-4}$ | 64 | 10 | cosine | 500 | BERT | Gruver et al. (2023) |
| D3PM-Baseline | 5 | $1\times10^{-4}$ | 64 | 10 | Sohl-Dickstein | 500 | ByteNet | Alamdari et al. (2024) |
| **D3PM** | 5 | $1\times10^{-4}$ | 64 | 10 | Sohl-Dickstein | 500 | ByteNet | Alamdari et al. (2024) |
| UDLM | 5 | $3\times10^{-5}$ | 64 | 2500 | loglinear | 500 | DiT | Schiff et al. (2024) |
| **MDLM** | 50 | $3\times10^{-4}$ | 64 | 2500 | loglinear | 500 | DiT | Schiff et al. (2024) |
| **ARLM** | 10 | $1\times10^{-4}$ | 32 | 10 | n/a | n/a | GPT-J | Nijkamp et al. (2023) |

### A.3.1 Diffusion over continuous space

Diffusion models construct samples by reversing a diffusion process that maps clean data points $\mathbf{x}_0$ to samples from a prior distribution $\pi(\mathbf{x})$. The forward process $(\mathbf{x}_0 \to \mathbf{x}_T)$ is composed of conditional distributions $p(\mathbf{x}_t|\mathbf{x}_{t-1})$, which admit closed-form expressions for the conditional distributions $p(\mathbf{x}_t|\mathbf{x}_0)$ and $p(\mathbf{x}_{t-1}|\mathbf{x}_t,\mathbf{x}_0)$. The reverse process $(\mathbf{x}_T \to \mathbf{x}_0)$ converts samples from the prior into samples from the learned data distribution $p_\theta(\mathbf{x}_0)$ by repeatedly predicting the denoised variable $\hat{\mathbf{x}}_0$ from noisy values $\mathbf{x}_t$, using the conditional distribution $p(\mathbf{x}_{t-1}|\mathbf{x}_t,\hat{\mathbf{x}}_0)$ to derive a transition distribution $p_\theta(\mathbf{x}_{t-1}|\mathbf{x}_t)$.

**Continuous noise forward process.** Similarly to Gruver et al. (2023), we define a protein sequence as $\mathbf{w} \in \mathcal{A}^L$, where $\mathcal{A}$ is the alphabet of amino acids and $L$ is the fixed length of the sequence. To learn a distribution $p(\mathbf{w})$, we first embed $\mathbf{w}$ into a continuous variable $\mathbf{x}_0$ using an embedding matrix $U_\theta$ or encoder from the ESM2 language model (Lin et al., 2023), transforming discrete tokens into a continuous latent space. Gaussian noise is then applied to this embedding space. The prior distribution is defined as:

$$\pi(\mathbf{x}) = \mathcal{N}(0, I), \tag{1}$$

while the forward process follows a Gaussian corruption schedule:

$$p(\mathbf{x}_t|\mathbf{x}_0) = \mathcal{N}(\sqrt{\bar{\alpha}_t}\mathbf{x}_0, (1-\bar{\alpha}_t)I), \quad \bar{\alpha}_t = \prod_{i=1}^{t}\alpha_i, \quad \alpha_t = 1 - \beta_t. \tag{2}$$

The variance schedule $\{\beta_t\}$ follows the cosine schedule proposed by Nichol & Dhariwal (2021), which is commonly used to stabilize training.

**Reverse process.** The reverse process aims to recover the original sequence by learning a function $p_\theta(\hat{\mathbf{w}}|\mathbf{x}_t, t)$ that predicts the sequence from noised points $\mathbf{x}_t$. This is done by minimizing the following objective:

$$L(\theta) = \mathbb{E}_{\mathbf{w}_0, t}\left[-\log p_\theta(\mathbf{w}_0|\mathbf{x}_t)\right], \quad \mathbf{x}_t \sim p(\mathbf{x}_t|\mathbf{x}_0 = U_\theta \mathbf{w}_0). \tag{3}$$

By learning $p_\theta(\hat{\mathbf{w}}|\mathbf{x}_t, t)$, we construct the reverse transition distribution:

$$p_\theta(\mathbf{x}_{t-1}|\mathbf{x}_t) = \sum_{\hat{\mathbf{w}}} p(\mathbf{x}_{t-1}|\mathbf{x}_t, \hat{\mathbf{x}}_0 = U_\theta \hat{\mathbf{w}}) p_\theta(\hat{\mathbf{w}}|\mathbf{x}_t, t), \tag{4}$$

where the posterior $p(\mathbf{x}_{t-1}|\mathbf{x}_t, \mathbf{x}_0)$ follows:

$$p(\mathbf{x}_{t-1}|\mathbf{x}_t, \mathbf{x}_0) = \mathcal{N}(\mathbf{x}_{t-1}; \mu_t, \sigma_t^2 I), \tag{5}$$

with mean $\mu_t$ and variance $\sigma_t^2$ given by:

$$\mu_t = \frac{\sqrt{\bar{\alpha}_{t-1}}\beta_t}{1 - \bar{\alpha}_t}\mathbf{x}_0 + \frac{\sqrt{\alpha_t}(1 - \bar{\alpha}_{t-1})}{1 - \bar{\alpha}_t}\mathbf{x}_t, \tag{6}$$

$$\sigma_t^2 = \frac{1 - \bar{\alpha}_{t-1}}{1 - \bar{\alpha}_t}\beta_t. \tag{7}$$

**Inference and sampling.** At inference time, the learned reverse process is used to generate protein sequences from the prior $\pi(x)$. This is done by iteratively sampling:

$$\mathbf{x}_{t-1} \sim p_\theta(\mathbf{x}_{t-1}|\mathbf{x}_t), \tag{8}$$

and then reconstructing $\mathbf{w}$ by sampling:

$$\mathbf{w} \sim p_\theta(\hat{\mathbf{w}}|\mathbf{x}_0). \tag{9}$$

This denoising process iteratively refines noisy embeddings back into structured sequences.

### A.3.2 Diffusion over discrete space.

Discrete diffusion models (Austin et al., 2021; Campbell et al., 2022; Lou et al., 2024) generate data in discrete spaces by reversing a predefined forward Markov process. Specifically, a family of distributions $p_t$ evolves according to the Markov chain

$$\frac{\mathrm{d}p_t}{\mathrm{d}t} = \boldsymbol{Q}_t p_t, \tag{10}$$

where $p_0 = p_{\text{data}}$ is the data distribution and $\boldsymbol{Q}_t \in \mathbb{R}^{N \times N}$ are predefined transition matrices.

This Markov process can be reversed with the help of a concrete score function, $s(\mathbf{x}, t) := [\frac{p_t(\tilde{\mathbf{x}})}{p_t(\mathbf{x})}]_{\tilde{\mathbf{x}} \neq \mathbf{x}}$, as its time reversal is given by

$$\frac{\mathrm{d}p_{T-t}}{\mathrm{d}t} = \bar{\boldsymbol{Q}}_{T-t} p_{T-t}, \tag{11}$$

where $\bar{\boldsymbol{Q}}_t[\tilde{\mathbf{x}}, \mathbf{x}] = s(\mathbf{x}, t)_{\tilde{x}} \boldsymbol{Q}_t[\mathbf{x}, \tilde{\mathbf{x}}]$ for $\tilde{\mathbf{x}} \neq \mathbf{x}$, and $\bar{\boldsymbol{Q}}_t[\mathbf{x}, \mathbf{x}] = -\sum_{\tilde{\mathbf{x}} \neq \mathbf{x}} \bar{\boldsymbol{Q}}_t[\tilde{\mathbf{x}}, \mathbf{x}]$. To generate data $\mathbf{x}_0 \sim p_{\text{data}}$, we start with sampling $\mathbf{x}_T$ from a uniform distribution and then evolve through Eq. 11 by the Euler method.

**Uniform discrete language models.** Both D3PM (Austin et al., 2021) and UDLM (Schiff et al., 2024) implement a uniform transition matrix $\boldsymbol{Q}_t = \frac{1}{N}\mathbf{1}\mathbf{1}^T - \boldsymbol{I}$. When $T \to \infty$, the probability distribution $p_T$ converges to a uniform distribution.

**Masked diffusion language models.** Masked diffusion language models (MDLM) (Sahoo et al., 2024) utilize an absorbing transition matrix $\boldsymbol{Q}_t$ that converts tokens in a sequence to [MASK] states. The corresponding transition matrix can be written as $\boldsymbol{Q}_t \in \mathbb{R}^{(N+1) \times (N+1)}$, $\boldsymbol{Q}_t = -\boldsymbol{I} + \mathbf{e}_{N+1}\mathbf{1}^T$. When $T \to \infty$, the limiting distribution $p_T$ converges to a completely masked sequence.

### A.3.3 Autoregressive language models.

In this work, we finetuned the ProGen2-small decoder-only transformer (151 million parameters) based on the code and parameters used in Yang et al. (2025a). Models were trained based on next token prediction and cross entropy loss. However, we did not use adapter layers, and we did not group batches based on sequence length. During inference from the autoregressive model, we used a temperature of 1 and a Top-$p$ value of 1.

## A.4 Steering methods

Table A2: **Summary of supervised value functions used to predict fitness in this work, to guide diffusion models.** All "classifiers" were trained as regressors to predict fitness. For DAPS methods, only clean data was used for training, whereas other classifiers are trained on clean and noised samples from various timesteps.

| Model | Guidance Strategy | Max Epochs | Learning Rate | Batch Size | Architecture | Hidden Dimension |
|---|---|---|---|---|---|---|
| Continuous Diffusion | CG | 1000 | $1 \times 10^{-3}$ | 128 | 4-layer MLP | 256 |
| | DAPS | 1000 | $1 \times 10^{-3}$ | 128 | 4-layer MLP | 256 |
| Continuous Diffusion | NOS | 100 | $1 \times 10^{-3}$ | 128 | 1-layer MLP | 256 |
| Discrete Diffusion | CG | 1000 | $1 \times 10^{-3}$ | 64 | 4-layer MLP | 64 |
| | DAPS | 200 | $1 \times 10^{-3}$ | 64 | 4-layer MLP | 64 |
| Discrete Diffusion | NOS | 200 | $3 \times 10^{-4}$ | 64 | linear layer | n/a |

### A.4.1 Classifier guidance

Classifier guidance (Song et al., 2021) is a technique used to steer samples generated by diffusion models toward desired attributes. The primary goal is to sample from a conditional distribution $p(\mathbf{x}|\mathbf{y})$, where $\mathbf{y}$ is a guiding signal of interest. In continuous space, this can be achieved by replacing the unconditional score function $\nabla_{\mathbf{x}_t} \log p_t(\mathbf{x}_t)$ at time $t$ by a conditional score function,

$$\nabla_{\mathbf{x}_t} \log p(\mathbf{x}_t|\mathbf{y}) = \nabla_{\mathbf{x}_t} \log p_t(\mathbf{x}_t) + \nabla_{\mathbf{x}_t} \log p_t(\mathbf{y}|\mathbf{x}_t) \tag{12}$$

To obtain the conditional score function, one only needs to train a time-dependent predictor, which predicts the probability of $p_t(\mathbf{y}|\mathbf{x}_t)$ given $\mathbf{x}_t$ and time $t$.

**Continuous guidance.** Classifier guidance modifies the reverse diffusion process to steer generated samples toward a desired property, represented by a conditioning variable $y$. The guided sampling process modifies the update rule for $\mathbf{x}_t$ by incorporating a classifier score $\nabla_{\mathbf{x}_t} \log p(\mathbf{y}|\mathbf{x}_t)$ into the model's learned score function based on the relation in Eq. 12. Following Song et al. (2021), the classifier guidance term modifies the predicted $\hat{\mathbf{x}}_0$ in the denoising process:

$$\hat{\mathbf{x}}_0 = \mathbf{x}_t + \sigma^2(s_\theta(\mathbf{x}_t, t) + \nabla_{\mathbf{x}_t} \log p(\mathbf{y}|\mathbf{x}_t)). \tag{13}$$

Since our diffusion model directly predicts logits rather than the score function $s_\theta(\mathbf{x}_t, t)$, adding classifier guidance requires modifying the predicted $\hat{\mathbf{x}}_0$.

Instead of predicting the score function explicitly, our model predicts logits over the vocabulary, from which the denoised representation $\hat{\mathbf{x}}_0$ is obtained. We modify $\hat{\mathbf{x}}_0$ by incorporating classifier gradients as follows:

- Compute the unmodified $\tilde{\mathbf{x}}_0$ using the model's predicted logits:

$$\tilde{\mathbf{x}}_0 = \sum_{\hat{\mathbf{w}}} p(\hat{\mathbf{w}}|\mathbf{x}_t, t) U_\theta \hat{\mathbf{w}} \tag{14}$$

Table A3: **Hyperparameters used to tune the guidance/steering process.** The **bolded** parameter was chosen as the ideal parameter for the iterative "Bayesian optimization" experiment (Fig. 6). Larger guidance parameter typically implements stronger guidance strength.

| Guidance Strategy | Hyperparameters |
|---|---|
| Continuous CG | $1/\beta = 64, 128, 256, 512, 1024$ |
| Discrete CG | $1/\beta = 1, 2.5, 6.25, \mathbf{15.625}, 39.0625$ |
| Continuous DAPS | $1/\beta = 0.25, 0.5, 1, 2, 4 \times 10^4$ |
| | $K = 50$ |
| | Euler method steps = 10 |
| | Langevin dynamics steps = 100 |
| Discrete DAPS | $1/\beta = 16, 32, 64, \mathbf{128}, 256$ |
| | $K = 50$ |
| | Euler method steps = 20 |
| | Metropolis Hastings steps = 1000 |
| Continuous NOS | $\lambda = 0.1, 1, 10, 100, \mathbf{1000}$ |
| | $\eta = 0.5, \mathbf{2}, 5$ |
| | $K = 5, \mathbf{10}$ |
| | optimizer = AdaGrad |
| Discrete NOS | $\lambda = 0.1, 1, 10, 100, 1000$ |
| | $\eta = 0.5, 2, 5$ |
| | $K = 5, 10$ |
| | optimizer = AdaGrad |
| DPO | $\beta = 0.02, 0.1, 0.5, \mathbf{2}, 4$ |
| | lr $= 1 \times 10^{-6}$ |
| | epochs = 5 |
| | batch size = 8 |

where $U_\theta$ is the embedding matrix mapping discrete tokens to continuous space.

- If a time-dependent classifier $f$ is available, compute the classifier guidance term:

$$\nabla_{\mathbf{x}_t} \log p(\mathbf{y}|\mathbf{x}_t) = \nabla_{\mathbf{x}_t} f(\mathbf{x}_t, t)/\beta. \tag{15}$$

- Modify $\tilde{\mathbf{x}}_0$ using the classifier gradient:

$$\hat{\mathbf{x}}_0 = \tilde{\mathbf{x}}_0 + \sigma^2 \nabla_{\mathbf{x}_t} \log p(\mathbf{y}|\mathbf{x}_t). \tag{16}$$

This allows the diffusion model to generate samples that are more likely to satisfy the desired condition $\mathbf{y}$.

Further details on training the classifier are provided in Table A2 and Table A3.

**Discrete guidance.**    Nisonoff et al. (2025) extend classifier guidance to discrete state-space diffusion models. In analogy to classifier guidance for continuous diffusion models, they modify the unconditional rate matrix $\bar{\mathbf{Q}}_t$ (as defined in Eq. 11) to be a conditional rate matrix $\mathbf{R}_t^{\mathbf{y}}$ with

$$\mathbf{R}_t^{\mathbf{y}}[\mathbf{x}, \tilde{\mathbf{x}}] = \frac{p(\mathbf{y}|\tilde{\mathbf{x}}, t)}{p(\mathbf{y}|\mathbf{x}, t)} \bar{\mathbf{Q}}_t[\mathbf{x}, \tilde{\mathbf{x}}], \ \forall \tilde{\mathbf{x}} \neq \mathbf{x}. \tag{17}$$

For classifier guidance on both continuous and discrete diffusion models, we train a time-dependent predictor (classifier) $f$ that predicts the fitness $\mathbf{y}$ given $\mathbf{x}_t$ at time $t$. We define $p(\mathbf{y}|\mathbf{x}) \propto \exp(f(\mathbf{x})/\beta)$, where $f(\cdot)$ is a surrogate predictor of the fitness, and $\beta$ is the guidance temperature and governs the strength of guidance. Therefore, $\nabla_{\mathbf{x}_t} \log p_t(\mathbf{y}|\mathbf{x}_t) = \frac{1}{\beta} \nabla_{\mathbf{x}_t} f(\mathbf{x}_t, t)$, and $\mathbf{R}_t^{\mathbf{y}}[\mathbf{x}, \tilde{\mathbf{x}}] = \exp\left(\frac{1}{\beta}\big(f(\tilde{\mathbf{x}}, t) - f(\tilde{\mathbf{x}}, t)\big)\right) \bar{\mathbf{Q}}_t[\mathbf{x}, \tilde{\mathbf{x}}]$.

To obtain a classifier $f$ for discrete diffusion models, we trained an MLP regressor to predict the fitness of a one-hot encoded sequence given $\mathbf{x}_t$ and uniformly random time $t \in [0, T]$. Further details are provided in Table A2 and Table A3.

### A.4.2 Posterior sampling

Another line of guidance work (Chung et al., 2023; Mardani et al., 2024; Zhang et al., 2025) focuses on drawing samples from the posterior distribution $p(\mathbf{x}|\mathbf{y}) \propto p(\mathbf{x})p(\mathbf{y}|\mathbf{x})$, where the prior distribution is modeled by a pretrained diffusion model. The conditional distribution $p(\mathbf{y}|\mathbf{x})$ can either be the likelihood function of a forward model (i.e., when $\mathbf{y}$ is an incomplete measurement of $\mathbf{x}$) or an exponential distribution with respect to a reward function (i.e., $p(\mathbf{y}|\mathbf{x}) \propto \exp(f(\mathbf{x})/\beta)$). The major difference between posterior sampling and classifier guidance is that it requires the reward function to be trained only on clean data $\mathbf{x}$.

While many works have studied posterior sampling in Euclidean space with continuous diffusion models, posterior sampling for discrete data has been less explored. We modified DAPS (Zhang et al., 2025) to enable diffusion posterior sampling in discrete-state spaces. Suppose $\mathbf{x}$ lies in a finite support $\mathcal{X}^D$, we follow the following steps:

- Initialize $\mathbf{x}_T \sim p_T(\mathbf{x}_T)$
- for $i = 1, \ldots, K$
    1. Sample $\hat{\mathbf{x}}_0^{(i)} \sim p(\mathbf{x}_0|\mathbf{x}_{t_{i-1}})$ by a discrete diffusion model.
    2. Run Metropolis Hastings to sample $\mathbf{x}_0^{(i)} \sim p(\mathbf{x}_0|\mathbf{x}_{t_{i-1}}, \mathbf{y})$ as defined in Eq. 18.
    3. Sample $\mathbf{x}_{t_i} \sim p(\mathbf{x}_{t_i}|\mathbf{x}_0)$ following the forward Markov process.
- Return $\mathbf{x}_K$.

Specifically, $t_0, t_1, \ldots, t_K$ are mono-decreasing time steps with $t_0 = T$ and $t_K \approx 0$. $p(\mathbf{x}_0|\mathbf{x}_t, \mathbf{y})$ is defined as

$$\begin{aligned}
p(\mathbf{x}_0|\mathbf{x}_t, \mathbf{y}) &\propto p(\mathbf{y}|\mathbf{x}_0)p(\mathbf{x}_0|\mathbf{x}_t) \\
&\approx p(\mathbf{y}|\mathbf{x}_0) \exp(-\|\mathbf{x}_0 - \hat{\mathbf{x}}_0(\mathbf{x}_t)\|_0/\sigma_t),
\end{aligned} \tag{18}$$

where $\hat{\mathbf{x}}_0(\mathbf{x}_t) \sim p(\mathbf{x}_0|\mathbf{x}_t)$ is a point estimate of the conditional distribution, and we approximate $p(\mathbf{x}_0|\mathbf{x}_t)$ by an exponential distribution over Hamming distance. Following Proposition 1 in Zhang et al. (2025), $\hat{\mathbf{x}}_0^{(i)}$, $\mathbf{x}_0^{(i)}$, and $\mathbf{x}_{t_i}$ converge to the posterior distribution as $t_i$ goes to 0.

For posterior sampling with DAPS, we obtained the value function $f$ using the same model architecture and training parameters as classifier guidance but only trained on clean data $\mathbf{x}$ (no noisy $\mathbf{x}_t$). We set $K = 50$ using the time scheduler for the original model. Further details are provided in Table A2 and Table A3.

### A.4.3 NOS

Diffusion optimized sampling (NOS) (Gruver et al., 2023) is a guidance method for both continuous and discrete diffusion models, which utilizes gradient information of the continuous latent representations of protein sequences. In pretrained discrete diffusion models, noisy sequences $\mathbf{w}_t$ always have a continuous embedding in the form of hidden states of the neural network. Specifically, the denoising model that predicts $\mathbf{w}_0$ from $\mathbf{w}_t$ can be written as $p_\theta(\mathbf{w}_0|g(\mathbf{w}_t), t)$, where $\mathbf{h}_t = g(\mathbf{w}_t)$ is a continuous hidden states of the model.

Instead of training a value function on discrete sequences $\mathbf{w}_t$, NOS proposes to train the value function on the hidden states $\mathbf{h}_t$. In each diffusion step, NOS samples from the posterior distribution,

$$p(\mathbf{w}_0|\mathbf{h}_t, \mathbf{y}) \propto p_\theta(\mathbf{w}_0|\mathbf{h}_t) \exp(f(\mathbf{h}_t)). \tag{19}$$

To sample from this distribution, NOS runs Langevin dynamics on $\mathbf{h}_t$, i.e.,

$$\mathbf{h}_t' \leftarrow \mathbf{h}_t' - \eta \nabla_{\mathbf{h}_t'}(\lambda D_{KL}(p_\theta(\mathbf{w}_0|\mathbf{h}_t')\|p_\theta(\mathbf{w}_0|\mathbf{h}_t)) - f(\mathbf{h}_t)) + \sqrt{2\eta\tau}\epsilon, \ \epsilon \sim \mathcal{N}(0, I). \tag{20}$$

After $K$ iterations, we denoise $\mathbf{w}_t$ following the guided hidden state, i.e., $p(\mathbf{w}_{t-1}|\mathbf{w}_t, \mathbf{y}) = p_\theta(\mathbf{w}_{t-1}|\mathbf{h}_t', t)$.

To train the value function used for guidance in NOS, following the method from Gruver et al. (2023), we trained a very shallow neural network on the final layer hidden embeddings of the diffusion model. Further details are provided in Table A2 and Table A3.

### A.4.4 Direct preference optimization

For DPO with language models, we used the weighted loss function from Widatalla et al. (2024) and Stocco et al. (2024) (Eq. 21). $\pi_\theta$ is the policy to be updated, $\pi_{\text{ref}}$ is the original model, and $\beta$ is a tunable parameter describing the extent of drift from the reference model. The loss therefore describes the cross entropy of the ratio $\beta \log \frac{\pi_\theta(\mathbf{x})}{\pi_{\text{ref}}(\mathbf{x})}$ and the fitness value $w$. Following Stocco et al. (2024), we calculated the ratio $r$ as the difference of the log likelihood of the sequence from the updated model minus the log likelihood of the reference model, and softmax was applied to all of the fitness values $w$. We used the default parameters from (Stocco et al., 2024) and tested increasing the learning rate to $10^{-4}$ but found that generation quality broke down above the levels used in Table A3 with finetuning for 5 epochs. We also tested ranked loss with other types of models, but the performance was similar.

$$L_{\text{DPO}_{\text{weighted}}}(\pi_\theta; \pi_{\text{ref}}) = -\mathbb{E}_D \sum_{k=1}^{K} w^k \left[ \beta \log \frac{\pi_\theta(x)}{\pi_{\text{ref}}(x)} - \log \sum_{j=k}^{K} \exp \left( \beta \log \frac{\pi_\theta(x)}{\pi_{\text{ref}}(x)} \right) \right] \qquad (21)$$

### A.5 Adaptive optimization algorithm

---
**Algorithm 1** Adaptive Optimization with Guided Generative Models

---
1: **Input:** Pretrained generative prior $p(\mathbf{x})$, initial empty labeled dataset $\mathcal{D}_0 = \emptyset$, number of rounds $T$, batch size $B$, ensemble size M
2: **for** $t = 1$ to $T$ **do**
3:      Initialize batch $\mathcal{X}_t \leftarrow \emptyset$
4:      **if** $t > 1$ **then** Train ensemble of value functions $\{f_{\theta_{t,m}}\}_{m=1}^{M}$ on $\mathcal{D}_{t-1}$
5:      **while** $|\mathcal{X}_t| < B$ **do**
6:          **if** $t > 1$ **then**
7:              Sample value function $f_\theta \sim \text{Uniform}(\{f_{\theta_{t,m}}\}_{m=1}^{M})$     ▷ Thompson-style sampling
8:              Sample sequence $\mathbf{x}_b \sim \text{GuidedSample}(p(\mathbf{x}), f_\theta, \text{GuidanceStrategy})$
9:          **else**
10:             Sample sequence $\mathbf{x}_b \sim \text{UnconditionalSample}(p(\mathbf{x}))$
11:          **end if**
12:          **if** $\mathbf{x}_b \notin \mathcal{D}_{t-1}$ **then** Add $\mathbf{x}_b$ to batch $\mathcal{X}_t$
13:      **end while**
14:      Evaluate true fitness $y_b = f_{\text{true}}(\mathbf{x}_b)$ for all $\mathbf{x}_b \in \mathcal{X}_t$
15:      Update dataset: $\mathcal{D}_t \leftarrow \mathcal{D}_{t-1} \cup \{(\mathbf{x}_b, y_b)\}_{b=1}^{B}$
16: **end for**
17: **Return:** Best observed sequence in $\mathcal{D}_T$

---

We used an ensemble size of $M = 10$ models, each trained with a different random initialization of neural network weights. In practice, to speed up sampling, we sampled ($B = 100$ samples)/($M = 10$ models) = 10 sequences in each GPU batch using the same Thompson-sampled value function, rather than using a GPU batch size of 1. Alternatively, for the Gaussian process model, we trained the model with the radial basis function kernel, and we sub-sampled the total amount of training pairs (when using noisy samples) to 5000 samples.

### A.6 Latent space Bayesian optimization with

We utilized the APEXGo codebase (Torres et al., 2024), a package for training generative variational autoencoders over peptide sequences and then optimizing those sequences with latent space Bayesian optimization to maximize certain properties. We used the training code out-of-the-box to train variational autoencoders over the same MSA sequences used to train priors for discrete diffusion models in SGPO. We trained until losses plateaued, to 542, 241, and 391 epochs for TrpB, CreiLOV, and GB1, respectively. Overall, the reconstruction losses were low, and generated sequences had low perplexity and high fitness, comparable to the generative models used in SGPO. We then used this latent space and the APEXGo optimization algorithm to maximize the fitness of sequences as measured by the oracle used in SGPO benchmarking. Specifically, in our configuration, we set

the number of initialization points to 100, the number of desired diverse solutions to 1, the max number of oracle calls to 800, and the batch size to 100–to mimic the iterative Bayesian optimization experiments performed in Fig. 6.

## A.7 Additional results

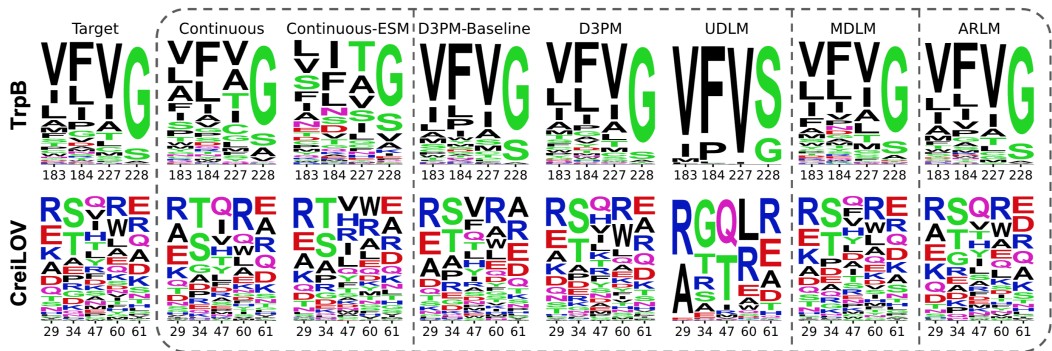

Figure A3: **The distributions of sequences sampled from pretrained generative priors largely match those of the target distribution.** The target distribution shows all sequences in the MSA, and the distributions of generative models are approximated by sampling 1000 sequences. Model definitions can be found in Table 3. The residues shown for TrpB are 4 out of 15 positions studied in the dataset (parent is VFVS), and 5 out of 119 residues for CreiLOV are shown as they correspond to those harboring favorable mutations in the original dataset (parent is AGQRD). Note that the target distribution for training the ARLM is slightly different than that shown here.

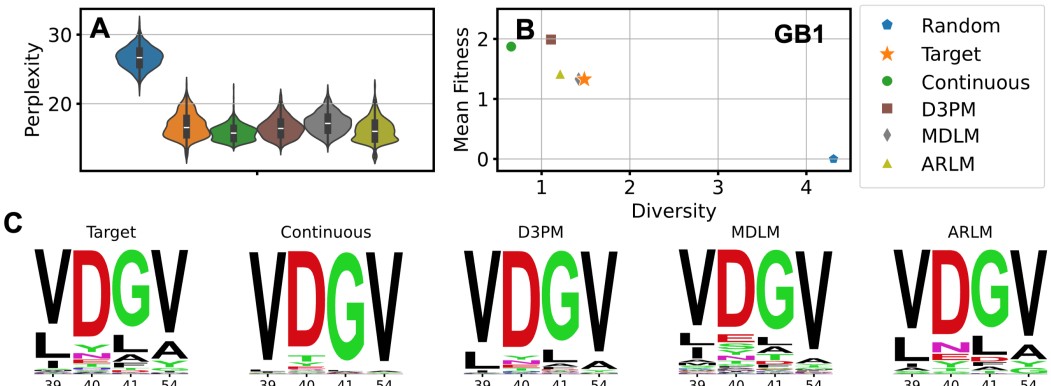

Figure A4: Additional results for GB1, corresponding to Fig. 4 and Fig. A3. The 4 positions shown correspond to the parent sequence of VDGV.

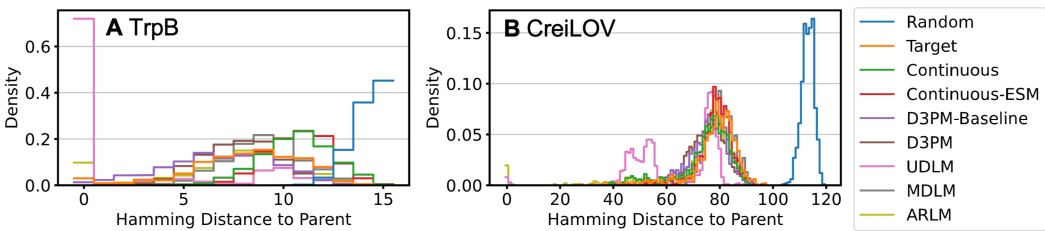

Figure A5: Generated sequences from pretrained priors are more similar to parent than random for (**A**) TrpB and (**B**) CreiLOV, measured by the Hamming (or edit) distance. UDLM models exhibit mode collapse onto consensus sequence(s) in the training distribution. The parent sequence refers to the starting sequence used to generate variants in the original protein fitness dataset.

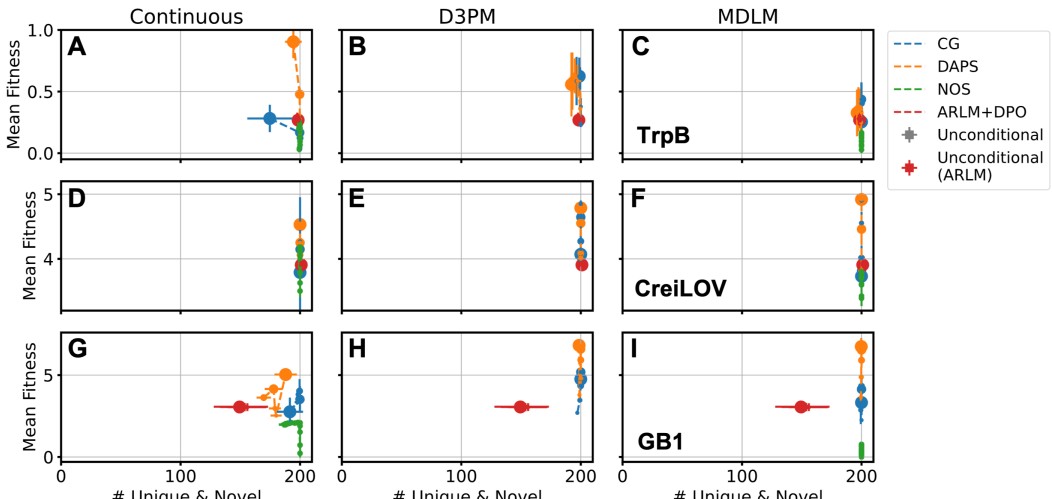

Figure A6: Pareto boundaries demonstrate the trade-off between generating sequences with high fitness and high diversity for TrpB (**A-C**), CreiLOV (**D-F**), and GB1 (**G-I**) – showing the same experiment as Fig. 5. Error bars show standard deviation. Mean fitness and diversity were calculated based on 200 generated samples, with diversity calculated as the total number of unique and novel (previously unseen) samples in the generated batch, out of 200. Larger circles indicate a stronger guidance strength, specified in Table. A3.

Table A4: Adaptive optimization with an ensemble of 10 value functions and Thompson sampling, compared to using a single model for guidance. Max fitness refers to the mean max fitness achieved at the end of the campaign using the same experimental setup as Fig. 6, over 5 different random initializations.

| Protein | Model | Guidance | Max Fitness (Ensemble) | Max Fitness (Single Model) |
|---|---|---|---|---|
| TrpB | D3PM | CG | **1.551** | 1.542 |
| | | DAPS | **1.595** | 1.568 |
| | MDLM | CG | **1.551** | 1.542 |
| | | DAPS | **1.595** | 1.568 |
| CreiLOV | D3PM | CG | **5.608** | 5.552 |
| | | DAPS | **5.522** | 5.520 |
| | MDLM | CG | **5.608** | 5.552 |
| | | DAPS | **5.530** | 5.520 |

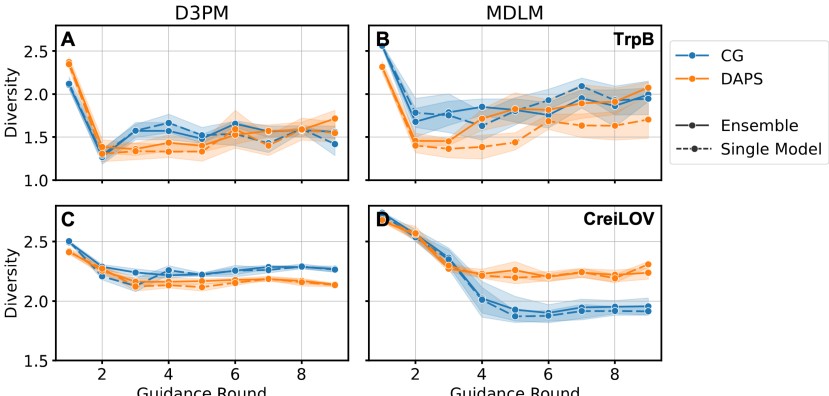

Figure A7: Diversity of generated sequences, measured by average Shannon entropy of mutated positions, during each round of guidance. Using an ensemble of value functions and Thompson sampling generally shows higher diversity than using a single model. Experimental setup is the same as Fig. 6, and experiments were repeated over 5 random initializations.

