# OpenReview forum: "Steering Generative Models with Experimental Data for Protein Fitness Optimization"
_NeurIPS.cc/2025/Conference — NeurIPS 2025 poster_

### Official Review · Reviewer_aBuC · 2025-06-30

**Clarity:** 2
**Significance:** 3
**Originality:** 2
**Rating:** 3
**Confidence:** 2

**Summary:**

The paper proposes a “steered-generation” pipeline for protein design, benchmarks three guidance strategies (CG, DAPS, NOS) across continuous and discrete diffusion models plus a DPO baseline, and shows that discrete diffusion with CG or DAPS works best on two protein datasets.

**Questions:**

Together with the above weaknesses, I have the following questions,

1. In Fig. 3, the authors points out many conbination among prior, learning strategy, and steering methods, but this study just discussed some of them. What about the remaining approaches?

2. In your experiments, why D3PM and MDLM models with the CG and DAPS guidance strategies achieved the best performance? Could you provide deeper explaination?

**Ethical Concerns:**

["NO or VERY MINOR ethics concerns only"]

**Final Justification:**

I think the authors' detailed responses and additional experiments have addressed my concerns.

**Limitations:**

The authors have discussed the limitations. The paper would benefit by providing the source code for reviewers to evaluate the reproducibility of the proposed method.

**Paper Formatting Concerns:**

No concerns.

**Quality:**

2

**Strengths And Weaknesses:**

**Strength:**
1. The study compares many guidance algorithms, diffusion architectures and an RL baseline under identical data budgets, filling a gap in prior work that typically evaluates a single combo.
2. The ensemble-Thompson strategy integrates uncertainty into generation and outperforms single-predictor guidance.

**Weakness:**
1. The paper is not well-organised, making it difficult to read. The language should be carefully polished.
2. In the experiment part, while the authors conducted many experiment on these two proteins, adding more datasets would strengthen the generality of the paper’s conclusions.
3. All results rely on an in-silico oracle, and no new wet-lab validation is provided.
4. Some citations were not be cited appropriately, such as EvoDiff in Table 3.

---

> ### Author Rebuttal · Authors · 2025-07-31
>
> We thank the reviewer for their feedback on our work. We respond to the specific comments below:
>
> ### **W1**: “The paper is not well-organised, making it difficult to read. The language should be carefully polished.”
>
> *Our Response:* We appreciate this feedback and will revise the manuscript to improve its clarity and organization. In particular, we plan to streamline the presentation of our experimental findings and rephrase several technical descriptions to ensure they are more accessible. If there are specific sections or passages that the reviewer found especially difficult to follow, we would be grateful for any pointers to help focus our revisions more effectively.
>
> ### **W2**: “Adding more datasets would strengthen the generality of the paper’s conclusions.”
>
> *Our Response:* Thank you for this suggestion. To strengthen the conclusions from our paper and better demonstrate the generalization ability of our method, we have added a third case study using the GB1 dataset [1], which is commonly used in the protein design literature. Unlike TrpB and CreiLOV, GB1 has a much smaller multiple sequence alignment (only 126 sequences), making it significantly more challenging in the low-data regime. Despite this, our methods continue to perform strongly on GB1. In particular:
>
> 1. DAPS consistently steers generation toward high-fitness variants, even with limited data, and does so while maintaining sequence diversity.
> 2. In adaptive optimization experiments, DAPS significantly outperforms CG and other baselines, reinforcing its robustness and effectiveness across different tasks.
>
> These new results further highlight the versatility of our framework and position DAPS as the strongest overall method in our evaluation. Figures 4–6 will be updated in the revision to include the GB1 results. For now, we provide a preview in tabular form below.
>
> **GB1 Results Table 1**
> |Model|Method|Guidance Strength|Mean Fitness (↑)|Diversity|# Unique & Novel|
> |:-----------|:---------|--------------------:|-------------------:|------------:|-------------------:|
> |Continuous|DAPS|2500|2.52908|0.320212|180|
> |Continuous|DAPS|5000|2.96662|0.307461|179|
> |Continuous|DAPS|10000|3.62737|0.276158|169.6|
> |Continuous|DAPS|20000|4.15179|0.267026|177.9|
> |Continuous|DAPS|40000|**5.0329**|0.29075|187.9|
> |D3PM|DAPS|16|3.79125|0.585082|198.9|
> |D3PM|DAPS|32|4.79|0.61515|200|
> |D3PM|DAPS|64|5.91924|0.629356|200|
> |D3PM|DAPS|128|6.59207|0.534547|199.9|
> |D3PM|DAPS|256|**6.81778**|0.436094|198.6|
> |MDLM|DAPS|16|3.65391|0.901675|199.5|
> |MDLM|DAPS|32|4.88479|0.803669|199.8|
> |MDLM|DAPS|64|5.90884|0.818706|200|
> |MDLM|DAPS|128|6.54008|0.65398|200|
> |MDLM|DAPS|256|**6.74237**|0.450862|199.8|
>
>
> **GB1 Results Table 2**
> |Method|Max Fitness Achieved(↑)|
> |:---------------------|---------------------------:|
> |DAPS (MDLM)|8.82489|
> |CG (MDLM)|7.78204|
> |Unconditional (MDLM)|4.54604|
> |Continuous+NOS|4.3317|
> |ARLM+DPO|4.30037|
> |Random|0.0169195|
>
> ### **W3**: “All results rely on an in-silico oracle, and no new wet-lab validation is provided.”
>
> *Our Response:* We agree that wet-lab validation is an important next step. However, the scope of our study, which evaluates a broad range of design choices, makes comprehensive experimental validation intractable. In practice, wet-lab experiments are typically designed to validate a single modeling hypothesis due to cost and throughput constraints [2]. That said, we are actively pursuing collaborations to evaluate top-performing methods from our benchmark in targeted experimental campaigns.
>
> ### **W4**: Additional citation.
>
> *Our Response:* We have now additionally included the EvoDiff citation in Table 3.
>
> ### **Q1**: “In Fig. 3, the authors points out many conbination among prior, learning strategy, and steering methods, but this study just discussed some of them. What about the remaining approaches?”
>
> *Our Response:* We elected to provide readers with a broad overview of the methods that fall under the umbrella of SGPO. In our study, we focused on a manageable number of plug-and-play guidance strategies, which have low computational costs and are easy and practical for real-world engineering campaigns, compared to fine-tuning approaches. Future work could implement and compare the other approaches.
>
> ### **Q2**: “In your experiments, why D3PM and MDLM models with the CG and DAPS guidance strategies achieved the best performance? Could you provide deeper explaination?”
>
> *Our Response:* We found that DAPS is the best steering strategy overall, consistently outperforming other guidance strategies such as CG and NOS. This may be because the classifier used in DAPS is only trained on clean data, whereas CG requires training the classifier on both clean and noisy data. We also found that the D3PM and MDLM models are better suited for discrete data, as they are able to capture the natural distribution of protein sequences, compared to the continuous model. Thus, for real world applications, we would suggest using DAPS + D3PM/MDLM.
>
> ### **L1**: “The paper would benefit by providing the source code for reviewers…”
>
> *Our Response:* We have developed a comprehensive and well-organized codebase that is already publicly available on a GitHub repository. However, due to this year’s rebuttal policy, we are unable to share the link at this stage.
>
> Please let us know if you have any further feedback or questions!
>
> ### Citations:
> [1] Olson, C. A.; Wu, N. C.; Sun, R. A Comprehensive Biophysical Description of Pairwise Epistasis throughout an Entire Protein Domain. Current Biology 2014, 24 (22), 2643–2651.
>
> [2] Yang, J.; Lal, R. G.; Bowden, J. C.; Astudillo, R.; Hameedi, M. A.; Kaur, S.; Hill, M.; Yue, Y.; Arnold, F. H. Active Learning-Assisted Directed Evolution. Nat Commun 2025, 16 (1), 714.

---

> > ### Comment · Reviewer_aBuC · 2025-08-05
> >
> > Thank you for your detailed responses and additional experiments. I believe some of my concerns have been addressed, and I will raise my score.

---

> > > ### Author Response · Authors · 2025-08-07
> > >
> > > Dear Reviewer ABuC,
> > >
> > > Thank you for your feedback during the review process and for increasing your score.
> > >
> > > Best regards,
> > > Authors

---

### Official Review · Reviewer_ZFcc · 2025-07-01

**Clarity:** 3
**Significance:** 2
**Originality:** 1
**Rating:** 3
**Confidence:** 4

**Summary:**

This paper tackles the task of protein fitness optimization, where one wants to generate realistic protein sequences that fulfill desired properties, i.e. fitness criteria. To this end, the paper investigates the performance of different guidance strategies (as well as direct preference optimization (DPO)) to sample different pre-trained generative models in a controlled manner, when only a small amount of experimental fitness data is available for guidance. The paper terms this "Steered Generation for Protein Optimization" (SGPO) and tests continuous-embedding diffusion models, uniform noise and masking-based discrete diffusion models as well as autoregressive models, using several different guidance methods (CG, DAPS, NOS; see paper for abbreviations and details). The paper quantitatively shows that for two protein optimization tasks, SGPO outperforms unconditional and random sequence sampling as well as autoregressive language models fine-tuned with DPO in certain situations. The paper quantifies this through diversity/fitness tradeoffs under different guidance strengths as well as a simulation of a real-world protein engineering setup, where samples are iteratively generated, then the value function is refined, and new samples are being generated.

**Questions:**

I do not have any further questions.

**Ethical Concerns:**

["NO or VERY MINOR ethics concerns only"]

**Final Justification:**

While I appreciate the authors' rebuttal, it does not fundamentally change my impression of the paper. There is very little methodological novelty in this work and this paper would seem better positioned at a venue specifically for protein design and not at a core machine learning conference. Please see my review for my other points of criticism.

That being said, this is a borderline decision and if others are more enthusiastic about the work, I would also support accepting the paper.

**Limitations:**

Limitations have been discussed appropriately.

**Paper Formatting Concerns:**

No concerns.

**Quality:**

3

**Strengths And Weaknesses:**

**Strengths:**
- The paper is well written and easy to follow.
- The paper provides a very thorough background section and discussion of various guidance methods, discrete diffusion models, as well as reinforcement learning-based methods.
- The paper tackles a practically relevant task, protein engineering, which has many important applications.

**Weaknesses:**
- There is little methodological novelty in the paper. The paper merely benchmarks established discrete generative modeling frameworks and guidance/steering/fine-tuning methods for protein fitness optimization.
- Since this is not a method development paper, but it rather focuses on benchmarking existing methods for application in real-world protein engineering, I would have hoped that the authors run very thorough experiments. However:
  - There is no wet-lab validation. This would have been appropriate in this case, I believe, because the main target audience of this paper is not machine learning researchers but protein engineers. The experiments leverage a computational oracle for evaluation and it is not clear whether this generalizes to real-world wet-lab performance.
  - The paper only studies two different protein optimization tasks and it is not entirely clear how generalizable the results are to other tasks. We can see that the results can vary quite a bit depending on the task. For instance, looking at Figure 6, we see that for TrpB and MDLM, the ARLM+DPO baseline performs as good as the other methods, whereas for CreiLOV, it performs worse than CG and DAPS. Hence, it is hard to make definite conclusions. Moreover, that these methods perform better than not doing any guidance or fine-tuning at all is a somewhat trivial finding.
  - Furthermore, as the authors pointed out, most methods come with various hyperparameters that could be investigated more deeply. While it is cumbersome and challenging to tune those hyperparameters, I would expect the authors to study method tuning in depth, and include thorough discussions and analyses on this in the main paper, in particular because the focus of the paper is to carefully benchmark these existing methods.

**Conclusion:** While the paper is generally well written and tackles a relevant problem, there is little methodological novelty and the benchmarking leaves some questions open; see above. Therefore, I am leaning towards rejection.

---

> ### Author Rebuttal · Authors · 2025-07-31
>
> We thank the reviewer for their feedback and for their support of our work. We respond to the specific comments below:
>
> ### **W1**: “There is little methodological novelty in the paper…”
>
> *Our Response:* We respectfully disagree with the claim that the paper lacks methodological novelty. While our focus is not on proposing a new generative model or steering algorithm, our main contribution lies in the conceptual unification and rigorous comparative evaluation of a broad class of steering techniques that have emerged in recent years but have largely been studied in isolation. Our SGPO framework draws meaningful connections between approaches from classifier guidance, Bayesian optimization, and reinforcement learning, and enables their systematic evaluation across multiple generative models and tasks.
>
> ### **W2**: “There is no wet-lab validation…”
>
> *Our Response:* We agree that wet-lab validation is an important next step. However, the scope of our study, which evaluates a broad range of design choices, makes comprehensive experimental validation intractable. In practice, wet-lab experiments are typically designed to validate a single modeling hypothesis due to cost and throughput constraints [1]. That said, we are actively pursuing collaborations to evaluate top-performing methods from our benchmark in targeted experimental campaigns.
>
> ### **W3**: “The paper only studies two different protein optimization tasks and it is not entirely clear how generalizable the results are to other tasks. ”
>
> *Our Response:* Prior to our study, it was not clear that DAPS + D3PM/MDLM would be such a strong model, consistently outperforming baselines such as NOS and DPO, which we believe is a valuable insight to both ML researchers looking to guide discrete diffusion models and protein engineers looking to choose a strategy for wet lab campaigns. To strengthen this conclusion, we have added a third case study using the GB1 dataset [2], which is commonly used in the protein design literature. Unlike TrpB and CreiLOV, GB1 has a much smaller multiple sequence alignment (only 126 sequences), making it significantly more challenging in the low-data regime. Despite this, our methods continue to perform strongly on GB1. In particular:
>
> 1. DAPS consistently steers generation toward high-fitness variants, even with limited data, and does so while maintaining sequence diversity.
> 2. In adaptive optimization experiments, DAPS significantly outperforms CG and other baselines, reinforcing its robustness and effectiveness across different tasks.
>
> These new results further highlight the versatility of our framework and position DAPS as the strongest overall method in our evaluation. Figures 4–6 will be updated in the revision to include the GB1 results. For now, we provide a preview in tabular form below.
>
> **GB1 Results Table 1**
> |Model|Method|Guidance Strength|Mean Fitness (↑)|Diversity|# Unique & Novel|
> |:-----------|:---------|--------------------:|-------------------:|------------:|-------------------:|
> |Continuous|DAPS|2500|2.52908|0.320212|180|
> |Continuous|DAPS|5000|2.96662|0.307461|179|
> |Continuous|DAPS|10000|3.62737|0.276158|169.6|
> |Continuous|DAPS|20000|4.15179|0.267026|177.9|
> |Continuous|DAPS|40000|**5.0329**|0.29075|187.9|
> |D3PM|DAPS|16|3.79125|0.585082|198.9|
> |D3PM|DAPS|32|4.79|0.61515|200|
> |D3PM|DAPS|64|5.91924|0.629356|200|
> |D3PM|DAPS|128|6.59207|0.534547|199.9|
> |D3PM|DAPS|256|**6.81778**|0.436094|198.6|
> |MDLM|DAPS|16|3.65391|0.901675|199.5|
> |MDLM|DAPS|32|4.88479|0.803669|199.8|
> |MDLM|DAPS|64|5.90884|0.818706|200|
> |MDLM|DAPS|128|6.54008|0.65398|200|
> |MDLM|DAPS|256|**6.74237**|0.450862|199.8|
>
>
> **GB1 Results Table 2**
> |Method|Max Fitness Achieved(↑)|
> |:---------------------|---------------------------:|
> |DAPS (MDLM)|8.82489|
> |CG (MDLM)|7.78204|
> |Unconditional (MDLM)|4.54604|
> |Continuous+NOS|4.3317|
> |ARLM+DPO|4.30037|
> |Random|0.0169195|
>
> ### **W4**: “While it is cumbersome and challenging to tune those hyperparameters, I would expect the authors to study method tuning in depth”
>
> *Our Response:* Thank you for this suggestion. We have now included a more comprehensive hyperparameter search over 30 combinations of NOS hyperparameters, following previous studies [3–4], across num_steps = 5, 10; step_size = 0.5, 2, 5; and stability_coefficient = 0.1, 1, 10, 100, 1000. Fig. 5 will be updated in the revision, but in the meantime we report the performance of the top 2 hyperparameter configurations in the table below, compared to DAPS with an optimized guidance strength. These new results corroborate our previous findings, confirming that DAPS methods consistently outperform NOS.
>
> **NOS Hyperparameters Results Table**
> |Protein|Model|Method|Mean Fitness (↑)|Diversity|nos_stability_coeff|nos_num_steps|nos_step_size|
> |:----------|:-----------|:---------|-------------------:|------------:|----------------------:|----------------:|----------------:|
> |TrpB|Continuous|DAPS|**0.904319**|1.2053||||
> |TrpB|Continuous|NOS|0.266629|1.80812|10|5|2|
> |TrpB|Continuous|NOS|0.259517|1.64129|100|10|5|
> |TrpB|MDLM|DAPS|**0.376086**|1.60617||||
> |TrpB|MDLM|NOS|0.16598|2.26534|1000|10|5|
> |TrpB|MDLM|NOS|0.161262|2.26718|0.1|5|0.5|
> |CreiLOV|Continuous|DAPS|**4.53023**|1.69757||||
> |CreiLOV|Continuous|NOS|4.19227|1.99188|100|10|2|
> |CreiLOV|Continuous|NOS|4.19123|1.98869|1000|10|2|
> |CreiLOV|MDLM|DAPS|**4.91902**|2.0076||||
> |CreiLOV|MDLM|NOS|3.81707|2.67316|1000|10|0.5|
> |CreiLOV|MDLM|NOS|3.81705|2.67689|1000|5|2|
>
> For DPO, we have also performed a more detailed hyperparameter search over the “ranked” and “weighted” loss variants, and this comparison will be included in the revision. Overall, we found that DPO still does not enable much steerability, with both loss forms performing similarly.
>
> Please let us know if you have any further feedback or questions!
> ### Citations:
> [1] Yang, J.; Lal, R. G.; Bowden, J. C.; Astudillo, R.; Hameedi, M. A.; Kaur, S.; Hill, M.; Yue, Y.; Arnold, F. H. Active Learning-Assisted Directed Evolution. Nat Commun 2025, 16 (1), 714.
>
> [2] Olson, C. A.; Wu, N. C.; Sun, R. A Comprehensive Biophysical Description of Pairwise Epistasis throughout an Entire Protein Domain. Current Biology 2014, 24 (22), 2643–2651.
>
> [3] Gruver, N.; Stanton, S.; Frey, N. C.; Rudner, T. G. J.; Hotzel, I.; Lafrance-Vanasse, J.; Rajpal, A.; Cho, K.; Wilson, A. G. Protein Design with Guided Discrete Diffusion. In Advances in Neural Information Processing Systems 36; 2023.
>
> [4] Schiff, Y.; Sahoo, S. S.; Phung, H.; Wang, G.; Boshar, S.; Dalla-torre, H.; Almeida, B. P. de; Rush, A.; Pierrot, T.; Kuleshov, V. Simple Guidance Mechanisms for Discrete Diffusion Models. In Thirteenth International Conference on Learning Representations, 2025.

---

> > ### Author Response · Authors · 2025-08-04
> >
> > Dear Reviewer ZFcc,
> >
> > As the discussion period draws to a close, we would greatly appreciate it if you could let us know whether our response has adequately addressed your concerns. If any questions remain, we would be happy to clarify them within the remaining time.
> >
> > We also encourage you to take a look at our recent exchange with Reviewer LY7c, particularly the discussion on SGPO as a unifying framework that connects multiple lines of research. We believe this discussion helps further clarify the novelty and contribution of our work, which was one of the key concerns in your review.
> >
> > If our responses have addressed your concerns, we kindly ask you to consider updating your score to reflect the improvements made in response to your helpful feedback.
> >
> > Thank you again for your time and thoughtful review.

---

> > > ### Comment · Reviewer_ZFcc · 2025-08-04
> > > **Thank you for rebuttal.**
> > >
> > > I would like to thank the authors for their rebuttal, which I have considered. I appreciate the additional experiments and explanations. Nonetheless, my overall impression of the paper has not substantially changed, so I maintain my rating.

---

> > > > ### Author Response · Authors · 2025-08-07
> > > >
> > > > Dear Reviewer ZFcc,
> > > >
> > > > Thank you again for your time and helpful feedback during the review process.
> > > >
> > > > Best regards,
> > > > Authors

---

### Official Review · Reviewer_LY7c · 2025-07-03

**Clarity:** 3
**Significance:** 3
**Originality:** 3
**Rating:** 5
**Confidence:** 5

**Summary:**

The paper considers the problem of protein fitness optimization, where the goal is to find a protein sequence that maximizes a certain property of interest. The paper provides a framework, termed Steered Generation for Protein Optimization (SGPO), which combines a pretrained generative model of protein sequences (capturing evolutionary information) with a small, iteratively updated dataset of labeled sequence-fitness pairs. The core idea is to use this limited labeled data to train a surrogate model for the fitness, which then "steers" the sampling process of the generative prior model to generate new, high-fitness candidate sequences. The paper empirically evaluates various generative priors (e.g., discrete diffusion models) and steering methods (e.g., classifier guidance, posterior sampling) in a simulated adaptive optimization loop.

**Questions:**

Please see Strengths And Weaknesses section.

**Ethical Concerns:**

["NO or VERY MINOR ethics concerns only"]

**Final Justification:**

The authors response and new results convinced me that this paper is a good contribution.

**Limitations:**

yes

**Quality:**

3

**Strengths And Weaknesses:**

- The problem studied in the paper is important for many real world applications.

- I especially commend the well-written and comprehensive python library that is released with the paper. Such artifacts can easily catalyze a lot of downstream work.

---

- The paper introduces "Steered Generation for Protein Optimization (SGPO)" as a "useful, general framework" (line 65) which suggests to the reader that this framework is being instantiated for the first time. I will categorize this as an instance of Bayesian optimization (BO) where the acquisition function optimization is guided by a generative model.  For example, it is mentioned in line 36 that “Firstly, SGPO utilizes labeled data, which is important for fitness goals that deviate from natural function” which is very natural and standard practice in a BO formulation. I assume the concept of adaptive optimization is probably more surprising for researchers working on developing generative models, who would typically focus on zero-shot or few-shot optimization, but this is completely standard practice for the BO community. Even more importantly, there is a large body of work combining BO with deep generative models now (please see [1-5]). Some of these papers are cited in the paper but not discussed meaningfully, mostly grouped with a big list of other methods. Please discuss this line of work and clearly differentiate your work within this context.

- I think the paper should invest more effort in making the baselines strong. The main contribution of the paper is a study of different methods and their combination. For example, statements like “NOS does not seem to allow for as much steerability, although we acknowledge that different hyperparameters could be tuned for NOS and DPO to potentially improve performance“ make the contribution a bit weaker. I believe precisely this is the type of paper where one should make an effort to ensure that the hyperparameters of the baselines are tuned properly. This way the community can learn a lot from the paper than its current form.

- Strong baselines like NOS[1], LOL-BO[2] were not included in the adaptive optimization part of the results. Please consider including them.

- It is mentioned that “Most existing approaches fail to incorporate principles from adaptive optimization” in line 58. I think this is not entirely correct. For example, LAMBO [Gruver et al] and many other ‘BO + generative models’ approaches do incorporate adaptive optimization principles.

- It is mentioned in line 150 in relation to  autoencoder style methods that “However, it can be difficult to ensure that decoded sequences are valid and biologically meaningful”. I think this is not true and in fact, the citation at the end of this sentence (Maus et al) works extremely well for the space of molecules with no such challenges.

- One of the main baselines in the paper is an autoregressive LLM finetuned with DPO. The choice of DPO in this setting seems a bit unclear since it converts absolute fitness values into pairwise preferences, which necessarily discards information available to the optimization process.

- I find the justification for TrPB dataset in Table 2 a bit surprising: “While the TrpB dataset has a lot more training labels, it may be more difficult to learn due to high amounts of epistatic effects between residues(non-additivity of mutation effects)”. I imagined these are precisely the kind of hard scenarios where we hope deep learning based generative models and supervised models should succeed. Otherwise, we could easily use simpler methods like a gaussian process with string kernels as surrogates and local greedy search/evolutionary search for selection of next candidates.

References

[1] Gruver, Nate, et al. "Protein design with guided discrete diffusion." Advances in neural information processing systems 36 (2023): 12489-12517.

[2] Maus, Natalie, et al. "Local latent space bayesian optimization over structured inputs." Advances in neural information processing systems 35 (2022): 34505-34518.

[3] Chu, Jaewon, et al. "Inversion-based Latent Bayesian Optimization." arXiv preprint arXiv:2411.05330 (2024).

[4] Kristiadi, Agustinus, et al. "A sober look at LLMs for material discovery: Are they actually good for Bayesian optimization over molecules?." arXiv preprint arXiv:2402.05015 (2024).

[5] Torres, Marcelo DT, et al. "A generative artificial intelligence approach for antibiotic optimization." bioRxiv (2024): 2024-11.

---
Overall, the big picture idea of the paper (adaptive optimization with deep generative models) is very interesting but execution of the paper is slightly underwhelming. I am happy to increase my score if all the questions are addressed properly.

---

> ### Author Rebuttal · Authors · 2025-07-31
>
> We thank the reviewer for their constructive feedback and for their support of our work. We respond to the specific comments below:
>
> ### **W1**: “Please discuss this [latent BO] line of work and clearly differentiate your work within this context.”
>
> *Our Response:* Thank you for raising this important point. We agree that adaptive optimization with labeled fitness data is standard in the Bayesian optimization (BO) community, and we appreciate your suggestion to clarify how our work relates to this literature.
>
> Our goal in introducing SGPO is to unify and systematize a broad class of emerging techniques, including classifier guidance, posterior sampling, and reinforcement learning, for guiding generative protein models with labeled data. Many of these approaches, particularly in the protein modeling community, are not typically framed in BO terms. SGPO provides a coherent perspective that connects these different lines of work.
>
> That said, we agree that SGPO is closely related to prior work combining BO with deep generative models that you have suggested, and we will revise the manuscript to clarify these connections. In particular, we will contrast SGPO with latent-space BO approaches and highlight key differences. SGPO offers greater flexibility by avoiding reliance on an explicit latent space, which enables the use of modern, more powerful generative models such as diffusion models and protein language models that are not easily accommodated by traditional latent BO pipelines. In response to your suggestion, we have also added APEX-GO (based on LOL-BO) from Torres et al. [1] as a representative baseline from the latent-space BO class of algorithms.
>
> The approach of Gruver et al. [2] can be viewed as an instance within our broader SGPO framework. SGPO supports a wider range of generative models and steering strategies, and this generality is reflected in the substantial empirical improvements achieved by several of our proposed methods over NOS, the guidance technique introduced in [2].
>
> ### **W2**: “I think the paper should invest more effort in making the baselines strong.”
>
> *Our Response:* Thank you for this suggestion. We have now included a more comprehensive hyperparameter search over 30 combinations of NOS hyperparameters following previous studies [2-3], across num_steps=5,10; step_size= 0.5,2,5; and stability_coefficient=0.1,1,10,100,1000. Fig. 5 will be updated in the revision, but in the meantime we report the performance of the top 2 ideal combinations of NOS hyperparameters in the table below, compared to the performance of DAPS with an optimized guidance strength. Ultimately, these new results corroborate previously reported results, confirming that DAPS methods perform better than NOS.
>
> **NOS Hyperparameters Results Table**
> |Protein|Model|Method|Mean Fitness (↑)|Diversity|nos_stability_coeff|nos_num_steps|nos_step_size|
> |:----------|:-----------|:---------|-------------------:|------------:|----------------------:|----------------:|----------------:|
> |TrpB|Continuous|DAPS|**0.904319**|1.2053||||
> |TrpB|Continuous|NOS|0.266629|1.80812|10|5|2|
> |TrpB|Continuous|NOS|0.259517|1.64129|100|10|5|
> |TrpB|MDLM|DAPS|**0.376086**|1.60617||||
> |TrpB|MDLM|NOS|0.16598|2.26534|1000|10|5|
> |TrpB|MDLM|NOS|0.161262|2.26718|0.1|5|0.5|
> |CreiLOV|Continuous|DAPS|**4.53023**|1.69757||||
> |CreiLOV|Continuous|NOS|4.19227|1.99188|100|10|2|
> |CreiLOV|Continuous|NOS|4.19123|1.98869|1000|10|2|
> |CreiLOV|MDLM|DAPS|**4.91902**|2.0076||||
> |CreiLOV|MDLM|NOS|3.81707|2.67316|1000|10|0.5|
> |CreiLOV|MDLM|NOS|3.81705|2.67689|1000|5|2|
>
> To better demonstrate the generalizability of our methods, we have added a third case study using the GB1 dataset [4], which is commonly used in the protein design literature. Unlike TrpB and CreiLOV, GB1 has a much smaller multiple sequence alignment (only 126 sequences), making it significantly more challenging in the low-data regime. Despite this, our methods continue to perform strongly on GB1. In particular:
>
> 1. DAPS consistently steers generation toward high-fitness variants, even with limited data, and does so while maintaining sequence diversity.
> 2. In adaptive optimization experiments, DAPS significantly outperforms CG and other baselines, reinforcing its robustness and effectiveness across different tasks.
>
> These new results further highlight the versatility of our framework and position DAPS as the strongest overall method in our evaluation. Figures 4–6 will be updated in the revision to include the GB1 results. For now, we provide a preview in tabular form below.
>
> **GB1 Results Table 1**
> |Model|Method|Guidance Strength|Mean Fitness (↑)|Diversity|# Unique & Novel|
> |:-----------|:---------|--------------------:|-------------------:|------------:|-------------------:|
> |Continuous|DAPS|2500|2.52908|0.320212|180|
> |Continuous|DAPS|5000|2.96662|0.307461|179|
> |Continuous|DAPS|10000|3.62737|0.276158|169.6|
> |Continuous|DAPS|20000|4.15179|0.267026|177.9|
> |Continuous|DAPS|40000|**5.0329**|0.29075|187.9|
> |D3PM|DAPS|16|3.79125|0.585082|198.9|
> |D3PM|DAPS|32|4.79|0.61515|200|
> |D3PM|DAPS|64|5.91924|0.629356|200|
> |D3PM|DAPS|128|6.59207|0.534547|199.9|
> |D3PM|DAPS|256|**6.81778**|0.436094|198.6|
> |MDLM|DAPS|16|3.65391|0.901675|199.5|
> |MDLM|DAPS|32|4.88479|0.803669|199.8|
> |MDLM|DAPS|64|5.90884|0.818706|200|
> |MDLM|DAPS|128|6.54008|0.65398|200|
> |MDLM|DAPS|256|**6.74237**|0.450862|199.8|
>
>
> **GB1 Results Table 2**
> |Method|Max Fitness Achieved(↑)|
> |:---------------------|---------------------------:|
> |DAPS (MDLM)|8.82489|
> |CG (MDLM)|7.78204|
> |Unconditional (MDLM)|4.54604|
> |Continuous+NOS|4.3317|
> |ARLM+DPO|4.30037|
> |Random|0.0169195|
>
> ### **W3**: “Strong baselines like NOS, LOL-BO were not included in the adaptive optimization part of the results. Please consider including them.”
>
> *Our Response:* We have now included NOS in the adaptive optimization experiments, and it continues to underperform relative to CG and DAPS. In addition, we have incorporated APEX-GO [1] (a recent protein-specific instantiation of LOL-BO) as a representative latent-space BO method. Initial results across all three datasets suggest that APEX-GO performs poorly in this setting (see the table below for GB1 adaptive optimization results). We hypothesize that this may be due to (i) the longer sequences used in our work compared to original LOL-BO studies, and (ii) the difficulty in calibrating the trust region in very low-data regimes with limited rounds of optimization. We will provide a more thorough discussion of these limitations in the revised manuscript and will also update Figure 6 to reflect these new results.
>
> **APEXGO Results Table**
> |# Sequences|Max Fitness DAPS (↑)|Max Fitness APEXGO (↑)|
> |--------------:|----------------------:|------------------------:|
> |100|1.95523|2.04042|
> |200|4.60165|2.7743|
> |300|6.37959|2.84413|
> |400|7.1716|2.83733|
> |500|7.59393|2.90728|
> |600|7.91073|2.81892|
> |700|8.11334|2.84832|
> |800|8.30358|2.89129|
> |900|8.46983|2.86372|
>
> ### **W4+5**:  Concerns about statements in Lines 58 and 150.
>
> *Our Response:* We have rewritten the specific sentence mentioned in line 58 and removed the sentence mentioned in line 150 to characterize existing work more accurately.
>
> ### **W6**: “The choice of DPO in this setting seems a bit unclear since it converts absolute fitness values into pairwise preference …”
>
> *Our Response:* We explored two different DPO implementations using the “ranked” and “weighted” loss forms as outlined in [5]. Namely, the weighted form is able to capture absolute fitness values. We have now performed a more detailed hyperparameter search over these two types of models, and this comparison will be included in the revision. Overall, we found that DPO still does not enable much steerability, with “ranked” and “weighted” loss forms performing similarly.
>
> ### **W7**: “I find the justification for TrPB dataset in Table 2 a bit surprising”
>
> *Our Response:* We appreciate the opportunity to clarify. Our intention was not to suggest that TrpB is uninteresting, but rather to note that the complexity of its fitness landscape (due to significant epistasis) poses a challenge despite the availability of many labeled sequences. We agree with the reviewer that such complexity is precisely what makes TrpB a valuable benchmark for testing generative modeling strategies. We have revised the phrasing accordingly.
>
> We thank the reviewer again for their constructive suggestions. We believe the revised manuscript will be significantly strengthened as a result of your feedback. Please let us know if there are any further points you'd like us to address.
> ### Citations:
> [1] Torres, M. D. T.; Zeng, Y.; Wan, F.; Maus, N.; Gardner, J.; De La Fuente-Nunez, C. A Generative Artificial Intelligence Approach for Antibiotic Optimization. bioRxiv, 2024.
>
> [2] Gruver, N.; Stanton, S.; Frey, N. C.; Rudner, T. G. J.; Hotzel, I.; Lafrance-Vanasse, J.; Rajpal, A.; Cho, K.; Wilson, A. G. Protein Design with Guided Discrete Diffusion. In Advances in Neural Information Processing Systems 36; 2023.
>
> [3] Schiff, Y.; Sahoo, S. S.; Phung, H.; Wang, G.; Boshar, S.; Dalla-torre, H.; Almeida, B. P. de; Rush, A.; Pierrot, T.; Kuleshov, V. Simple Guidance Mechanisms for Discrete Diffusion Models. In Thirteenth International Conference on Learning Representations, 2025.
>
> [4] Olson, C. A.; Wu, N. C.; Sun, R. A Comprehensive Biophysical Description of Pairwise Epistasis throughout an Entire Protein Domain. Current Biology 2014, 24 (22), 2643–2651.
>
> [5] Stocco, F.; Artigues-Lleixà, M.; Hunklinger, A.; Widatalla, T.; Güell, M.; Ferruz, N. Guiding Generative Protein Language Models with Reinforcement Learning. arXiv 2025.

---

> > ### Comment · Reviewer_LY7c · 2025-08-02
> > **Thanks for response**
> >
> > Thank you to the authors for taking the time to respond to my comments. Overall, I am happy to increase my score towards positive now. I have few followup comments if the authors would kindly consider.
> >
> > - Re: **“SGPO provides a coherent perspective that connects these different lines of work.”** I think this is a strong point of the paper and a valuable contribution to the community.
> >
> > - Thanks for adding the new baseline of Torres et al [1]. This is a stronger baseline and great to see the results still hold up well.
> >
> > - Re: **“we agree that SGPO is closely related to prior work combining BO with deep generative models that you have suggested, and we will revise the manuscript to clarify these connections.”**, I request the authors to spend more time and dig a bit deeper on this discussion since it will significantly strengthen the paper. I believe the current response that “SGPO offers greater flexibility” might not be good enough. Take NOS [Gruver et al] for example, which uses a diffusion model but still works in the latent space (hidden embeddings of the diffusion model). Even for complex domains like video generation, latent diffusion models seem to be more effective in performance (see Generative modelling in latent space by Sander Dielman). Explicit latent space is actually quite nice since it gives us a well specified space to perform any form of acquisition or scoring optimization. I acknowledge that there are hyperparameters that need to be chosen like the boundary of the latent dimensions but that is true for steering style methods too.
> >
> > - Thanks for adding the new benchmark function as well. I was surprised to see no discussion of protein design benchmarks like proteingym. I am not asking for new results but just curious if there is no standardized benchmark yet in protein sequence design? That seems odd given how active this area is, but maybe I'm missing something.
> >
> > [1] Notin, Pascal, et al. "Proteingym: Large-scale benchmarks for protein fitness prediction and design." Advances in Neural Information Processing Systems 36 (2023): 64331-64379.

---

> ### Author Response · Authors · 2025-08-04
>
> First, we thank the reviewer again for their thoughtful feedback and for increasing their score. We address the follow-up comments below.
>
> ### “I think this is a strong point of the paper and a valuable contribution to the community.”
>
> *Our response:* Thank you for this generous comment. We are glad you found the unifying perspective of SGPO to be a valuable contribution. We will emphasize this aspect more clearly in the revision and further elaborate on its conceptual connections to prior work.
>
> ### “Thanks for adding the new baseline of Torres et al [1]. This is a stronger baseline and great to see the results still hold up well.”
>
> *Our response:*  We appreciate the suggestion to include baselines from the latent space BO literature. We are pleased that the results continue to support the strength of SGPO methods and will highlight this comparison in the revised manuscript.
>
> ### “I request the authors to spend more time and dig a bit deeper on this discussion since it will significantly strengthen the paper.”
> *Our response:* Thank you for encouraging us to clarify the relationship between SGPO and prior work on Bayesian optimization with generative models. We agree that making these connections more explicit will significantly strengthen the paper, and we will revise the manuscript accordingly.
>
> While both SGPO and latent space BO aim to leverage generative models for guided sequence design, they differ in how the optimization process is structured. Latent space BO methods typically define an explicit optimization problem in a learned latent space: a surrogate model is trained to predict outcomes based on latent codes, and acquisition functions are optimized to select promising codes, which are then decoded. This approach provides a well-defined optimization landscape but relies heavily on the structure of the latent space learned during generative model training. In protein design, such latent spaces are often shaped by distributions over natural sequences, which can limit extrapolation to high-fitness but unnatural variants. Prior work (e.g., Maus et al., 2022) has shown that latent BO can underperform when the latent space is not well aligned with the property being optimized. One possible remedy is to fine-tune the generative model using objective values in an end-to-end fashion, but this can be computationally expensive and may introduce additional challenges, such as overfitting or mode collapse.
>
> SGPO, by contrast, steers generation directly in sequence space using posterior-guided sampling. This avoids the need for a fixed latent space and supports a broader range of generative models, including diffusion models and protein language models without encoder-decoder architectures. It also enables plug-and-play use of scoring models trained solely on labeled data, allowing for greater adaptability to new objectives. That said, we believe there is promising future work in incorporating ideas from latent BO, such as local surrogate modeling, within the SGPO framework.
>
> The case of NOS (Gruver et al., 2023) is particularly interesting. Although NOS perturbs internal activations of the denoising network—effectively manipulating a latent representation—it does not optimize an acquisition function in this space as traditional latent BO methods do. Instead, its mechanism resembles classifier guidance, modifying the generative trajectory during sampling. For this reason, we view NOS as conceptually closer to SGPO than to latent BO. Nevertheless, our experiments show that NOS underperforms relative to SGPO methods like DAPS and CG. We believe that DAPS’s strong performance can be explained by the fact that DAPS leverages a classifier trained only on clean data and applies input-level guidance.
>
> We will revise the manuscript to incorporate this expanded discussion, clarify conceptual distinctions, and cite the relevant literature.
>
> ### “I am not asking for new results but just curious if there is no standardized benchmark yet in protein sequence design?”
> *Our response:* Indeed, there is currently no standardized oracle or dataset for sequence-level protein design over a large combinatorial space, where the goal is to identify combinations of multiple mutations—not just predict single mutation effects, which is the focus of ProteinGym. While some prior studies have used datasets like AAV or GFP, the optimization tasks and oracles used are not standardized, and none explicitly leverage both natural sequence priors and labeled fitness data, as we do in our study. We chose to focus on TrpB and CreiLOV because they are newer and more challenging benchmarks. That said, we agree that developing a more comprehensive and standardized benchmark for protein sequence design is an important direction for future work.

---

> > ### Comment · Reviewer_LY7c · 2025-08-04
> >
> > Thanks for the answers. I am happy with the response and I will also raise few sub-scores.

---

> > > ### Author Response · Authors · 2025-08-07
> > >
> > > Dear Reviewer LY7c,
> > >
> > > Thank you again for your helpful feedback during the review process and for increasing your scores.
> > >
> > > Best regards,
> > > Authors

---

### Official Review · Reviewer_kDSu · 2025-07-03

**Clarity:** 3
**Significance:** 3
**Originality:** 3
**Rating:** 5
**Confidence:** 2

**Summary:**

This paper introduces SGPO (Steered Generation for Protein Optimization), a framework for finding protein sequences with desired properties. The method pretrains generative models on naturally occurring protein sequences, then steers generation using limited experimental fitness data. The authors compare generative models and steering strategies, finding that plug-and-play guidance with diffusion models outperforms reinforcement learning approaches like DPO when data is scarce. They also present an adaptive optimization strategy using ensemble fitness predictors to efficiently explore protein sequence space and identify high-fitness variants.

**Questions:**

- Could the authors provide additional guidance on how practitioners should select or tune the guidance strength hyperparameter in real-world, held-out experimental campaigns? Are there any unsupervised or data-driven heuristics that might mitigate this sensitivity?

**Ethical Concerns:**

["NO or VERY MINOR ethics concerns only"]

**Final Justification:**

The authors' rebuttal has addressed the generalizability and hyperparameter selection issue.

**Limitations:**

yes

**Quality:**

3

**Strengths And Weaknesses:**

Strengths：
- The paper offers an extensive comparison of generative modeling and steering strategies, including multiple types of diffusion models (continuous, uniform discrete, absorbing-state discrete), plug-and-play guidance methods, and RL finetuning baselines.
- SGPO is contextualized as a unifying framework addressing the limitations of zero-shot, supervised, and prior-only methods. The approach is directly relevant to core challenges in protein engineering, such as efficiency in low-data regimes and scalability to large design spaces.

Weaknesses:
- The main experiments are conducted on just TrpB and CreiLOV, both of which are relatively "tractable" due to large multiple sequence alignments. While the justification is reasonable, this limits the assessment of broader generalizability to more challenging or less-studied protein families. Results on a more diverse set of proteins or harder fitness objectives would significantly strengthen impact.
- While guidance-based methods require tuning only a single parameter, the results show significant performance dependence on this hyperparameter. In real optimization campaigns, it is unclear how users would choose this value in the absence of held-out ground truth.

---

> ### Author Rebuttal · Authors · 2025-07-31
>
> We thank the reviewer for their constructive feedback and their support of our work. Below we address the specific concerns raised.
>
> ### **W1**: “Results on a more diverse set of proteins or harder fitness objectives would significantly strengthen impact.”
>
> *Our response:* We thank the reviewer for this suggestion. To better demonstrate the generalizability of our methods, we have added a third case study using the GB1 dataset [1], which is commonly used in the protein design literature. Unlike TrpB and CreiLOV, GB1 has a much smaller multiple sequence alignment (only 126 sequences), making it significantly more challenging in the low-data regime. Despite this, our methods continue to perform strongly on GB1. In particular:
>
> 1. DAPS consistently steers generation toward high-fitness variants, even with limited data, and does so while maintaining sequence diversity.
> 2. In adaptive optimization experiments, DAPS significantly outperforms CG and other baselines, reinforcing its robustness and effectiveness across different tasks.
>
> These new results further highlight the versatility of our framework and position DAPS as the strongest overall method in our evaluation. Figures 4–6 will be updated in the revision to include the GB1 results. For now, we provide a preview in tabular form below.
>
> **GB1 Results Table 1**
> | Model      | Method   |   Guidance Strength |   Mean Fitness (↑) |   Diversity |   # Unique & Novel |
> |:-----------|:---------|--------------------:|-------------------:|------------:|-------------------:|
> | Continuous | DAPS     |                2500 |            2.52908 |    0.320212 |              180   |
> | Continuous | DAPS     |                5000 |            2.96662 |    0.307461 |              179   |
> | Continuous | DAPS     |               10000 |            3.62737 |    0.276158 |              169.6 |
> | Continuous | DAPS     |               20000 |            4.15179 |    0.267026 |              177.9 |
> | Continuous | DAPS     |               40000 |            **5.0329**  |    0.29075  |              187.9 |
> | D3PM       | DAPS     |                  16 |            3.79125 |    0.585082 |              198.9 |
> | D3PM       | DAPS     |                  32 |            4.79    |    0.61515  |              200   |
> | D3PM       | DAPS     |                  64 |            5.91924 |    0.629356 |              200   |
> | D3PM       | DAPS     |                 128 |            6.59207 |    0.534547 |              199.9 |
> | D3PM       | DAPS     |                 256 |            **6.81778** |    0.436094 |              198.6 |
> | MDLM       | DAPS     |                  16 |            3.65391 |    0.901675 |              199.5 |
> | MDLM       | DAPS     |                  32 |            4.88479 |    0.803669 |              199.8 |
> | MDLM       | DAPS     |                  64 |            5.90884 |    0.818706 |              200   |
> | MDLM       | DAPS     |                 128 |            6.54008 |    0.65398  |              200   |
> | MDLM       | DAPS     |                 256 |            **6.74237** |    0.450862 |              199.8 |
>
> **GB1 Results Table 2**
> | Method               |   Max Fitness Achieved (↑) |
> |:---------------------|---------------------------:|
> | DAPS (MDLM)          |                  8.82489   |
> | CG (MDLM)            |                  7.78204   |
> | Unconditional (MDLM) |                  4.54604   |
> | Continuous+NOS       |                  4.3317    |
> | ARLM+DPO             |                  4.30037   |
> | Random               |                  0.0169195 |
>
> ### **W2/Q1**: “Could the authors provide additional guidance on how practitioners should select or tune the guidance strength hyperparameter in real-world, held-out experimental campaigns?”
>
> *Our Response:* Thank you for raising this important point. We would first like to emphasize that the methods explored in our study, such as CG and DAPS, are intentionally designed to be simple to tune, with only a single steering hyperparameter. This is in contrast to methods like NOS and DPO, which involve more complex and sensitive hyperparameter choices. We will add the following sentence to the revision to offer more concrete guidance:
>
> “In real-world engineering scenarios, even in the absence of ground truth fitness labels, one practical approach to selecting the guidance strength is to scan over values and choose the highest setting for which the N generated sequences remain unique and novel relative to previously measured sequences, where N corresponds to the screening throughput available for the next round.”
>
> This ensures that the chosen guidance strength balances optimization with diversity and remains actionable under practical experimental constraints.
>
> Please let us know if you have any further feedback or questions!
>
> ### Citations:
>
> [1] Olson, C. A.; Wu, N. C.; Sun, R. A Comprehensive Biophysical Description of Pairwise Epistasis throughout an Entire Protein Domain. Current Biology 2014, 24 (22), 2643–2651.

---

> > ### Author Response · Authors · 2025-08-04
> >
> > Dear Reviewer kDSu,
> >
> > As the end of the discussion period approaches, we would greatly appreciate it if you could confirm whether our response has adequately addressed your concerns.
> >
> > If your concerns have been resolved, we kindly ask you to consider updating your score, as this would reflect the improvements made based on your valuable feedback.
> >
> > Thank you again for your time and efforts in reviewing our manuscript.

---

> > > ### Comment · Reviewer_kDSu · 2025-08-07
> > >
> > > Thank you for your reply. Most of the concerns are addressed. So I have raised the score accordingly.

---

> > > > ### Author Response · Authors · 2025-08-07
> > > >
> > > > Dear Reviewer kDSu,
> > > >
> > > > Thank you again for your helpful feedback during the review process and for increasing your score. We noticed that your "rating" is still visible and hasn't been updated – could you kindly double check that you have successfully updated it?
> > > >
> > > > Best regards, Authors

---

### Note · Authors · 2025-08-11

Dear Reviewers,

We would like to thank you again for your engagement during the discussion period. We are glad that our responses addressed your primary concerns, leading three of the four reviewers to raise their scores. Your feedback will be carefully incorporated into the revised manuscript.

To Reviewer kDSu, we kindly note that your score appears not to have been updated yet, and we would greatly appreciate it if you could do so at your convenience.

Thank you again for your contributions to the review process and for your support of our work.

Sincerely,

The Authors

---

### Decision · Program_Chairs · 2025-09-17

**Decision:**

Accept (poster)

**Comment:**

This paper proposes SGPO, a method to generate protein sequences with desired properties. The method pretrains generative models on naturally occurring protein sequences, then steers generation using limited experimental fitness data.

Strengths：
- The overall plug-and-play nature is qutie appealing with small experimental data.
- The paper offers an extensive comparison of generative modeling and steering strategies


Weaknesses:
- The main experiments are conducted on just TrpB and CreiLOV, both of which are relatively "tractable" due to large multiple sequence alignments. While the justification is reasonable, this limits the assessment of broader generalizability to more challenging or less-studied protein families. Results on a more diverse set of proteins or harder fitness objectives would significantly strengthen impact.
- the method overly relies on one hyperparameter to tune.
- the method itself lacks novelty, but is rather a Bayesian optimization technique.